# Understanding and Improving Training-free Loss-based Diffusion Guidance

**Yifei Shen**[1]* **Xinyang Jiang**[1] **Yifan Yang**[1] **Yezhen Wang**[2] **Dongqi Han**[1] **Dongsheng Li**[1]
[1]Microsoft Research Asia  [2]National University of Singapore

## Abstract

Adding additional guidance to pretrained diffusion models has become an increasingly popular research area, with extensive applications in computer vision, reinforcement learning, and AI for science. Recently, several studies have proposed training-free loss-based guidance by using off-the-shelf networks pretrained on clean images. This approach enables zero-shot conditional generation for universal control formats, which appears to offer a free lunch in diffusion guidance. In this paper, we aim to develop a deeper understanding of training-free guidance, as well as overcome its limitations. We offer a theoretical analysis that supports training-free guidance from the perspective of optimization, distinguishing it from classifier-based (or classifier-free) guidance. To elucidate their drawbacks, we theoretically demonstrate that training-free guidance is more susceptible to misaligned gradients and exhibits slower convergence rates compared to classifier guidance. We then introduce a collection of techniques designed to overcome the limitations, accompanied by theoretical rationale and empirical evidence. Our experiments in image and motion generation confirm the efficacy of these techniques.

## 1 Introduction

Diffusion models represent a class of powerful deep generative models that have recently broken the long-standing dominance of generative adversarial networks (GANs) [8]. These models have demonstrated remarkable success in a variety of domains, including the generation of images and videos in computer vision [33, 3], the synthesis of molecules and proteins in computational biology [20, 52], as well as the creation of trajectories and actions in the field of reinforcement learning (RL) [21].

One critical area of research in the field of diffusion models involves enhancing controllability, such as pose manipulation in image diffusion [50], modulation of quantum properties in molecule diffusion [20], and direction of goal-oriented actions in RL diffusion [23]. The predominant techniques for exerting control over diffusion models include classifier guidance and classifier-free guidance. Classifier guidance involves training a time-dependent classifier to map a noisy image, denoted as $x_t$, to a specific condition $y$, and then employing the classifier's gradient to influence each step of the diffusion process [8]. Conversely, classifier-free guidance bypasses the need for a classifier by training an additional diffusion model conditioned on $y$ [18]. However, both approaches necessitate extra training to integrate the conditions. Moreover, their efficacy is often constrained when the data-condition pairs are limited and typically lack the zero-shot generalization capability.

Recently, several studies [2, 49, 37] have introduced training-free guidance that builds upon the concept of classifier guidance. These models eschew the need for training a classifier on noisy images; instead, they estimate the clean image from its noisy counterpart using Tweedie's formula and then employ pretrained networks, designed for clean images, to guide the diffusion process. Given that

---

*Contact `yifeishen@microsoft.com`

38th Conference on Neural Information Processing Systems (NeurIPS 2024).

checkpoints for these networks pretrained on clean images are widely accessible online, this form of guidance can be executed in a zero-shot manner. A unique advantage of training-free guidance is that it can be applied to universal control formats, such as style, layout, and FaceID [2, 49, 37] without any additional training efforts. Furthermore, these algorithms have been successfully applied to offline reinforcement learning, enabling agents to achieve novel goals not previously encountered during training. In contrast to classifier guidance and classifier-free guidance, it is proved in Appendix E of [27] that training-free guidance does not offer an approximation to the exact conditional energy. Therefore, from a theoretical perspective, it is intriguing to understand how and when these methods succeed or fail. From an empirical standpoint, it is crucial to develop algorithms that can address and overcome these limitations.

This paper seeks to deepen the understanding of training-free guidance by examining its mechanisms and inherent limitations, as well as overcoming these limitations. Specifically, our major contributions can be summarized as follows:

- **How does training-free guidance work?** Although exact conditional energy is difficult to approximate in a training-free manner, from the optimization standpoint, we show that training-free guidance can effectively decrease the guidance loss function. The optimization perspective clarifies the mystery of why the guidance weights should be meticulously designed in relation to the guidance function and diffusion time, as observed in [49].

- **When does training-free guidance not work?** We theoretically identify the susceptibility of training-free guidance to misaligned gradient issues and slower convergence rates. We attribute these challenges to a decrease in the smoothness of the guidance network in contrast to the classifier guidance.

- **Improving training-free guidance:** We introduce random augmentation to alleviate the misaligned gradient and Polyak step size scheduling to improve convergence. The efficacy of these methods is empirically confirmed across various diffusion models (i.e., image diffusion and motion diffusion) and under multiple conditions (i.e., segmentation, sketch, text, object avoidance, and targeting)[2].

## 2 Preliminaries

### 2.1 Diffusion Models

Diffusion models are characterized by forward and reverse processes. The forward process, occurring over a time interval from $0$ to $T$, incrementally transforms an image into Gaussian noise. On the contrary, the reverse process, from $T$ back to $0$, reconstructs the image from the noise. Let $\boldsymbol{x}_t$ represent the state of the data point at time $t$; the forward process systematically introduces noise to the data by following a predefined noise schedule given by $\boldsymbol{x}_t = \sqrt{\alpha_t}\boldsymbol{x}_0 + \sigma_t\boldsymbol{\epsilon}_t$, where $\alpha_t \in [0, 1]$ is monotonically decreasing with $t$, $\sigma_t = \sqrt{1 - \alpha_t}$, and $\boldsymbol{\epsilon}_t \sim \mathcal{N}(0, \boldsymbol{I})$ is random noise. Diffusion models use a neural network to learn the noise at each step:

$$\min_{\theta} \mathbb{E}_{\boldsymbol{x}_t, \boldsymbol{\epsilon}, t}[\|\boldsymbol{\epsilon}_\theta(\boldsymbol{x}_t, t) - \boldsymbol{\epsilon}_t\|_2^2] = \min_{\theta} \mathbb{E}_{\boldsymbol{x}_t, \boldsymbol{\epsilon}, t}[\|\boldsymbol{\epsilon}_\theta(\boldsymbol{x}_t, t) + \sigma_t\nabla_{\boldsymbol{x}_t} \log p_t(\boldsymbol{x}_t)\|_2^2],$$

where $p_t(\boldsymbol{x}_t)$ is the distribution of $\boldsymbol{x}_t$. The reverse process is obtained by the following ODE:

$$\frac{\mathrm{d}\boldsymbol{x}_t}{\mathrm{d}t} = f(t)\boldsymbol{x}_t - \frac{g^2(t)}{2}\nabla_{\boldsymbol{x}_t} \log p_t(\boldsymbol{x}_t) = f(t)\boldsymbol{x}_t + \frac{g^2(t)}{2\sigma_t}\boldsymbol{\epsilon}_\theta(\boldsymbol{x}_t, t), \tag{1}$$

where $f(t) = \frac{\mathrm{d}\log\sqrt{\alpha_t}}{\mathrm{d}t}$, $g^2(t) = \frac{\mathrm{d}\sigma_t^2}{\mathrm{d}t} - 2\frac{\mathrm{d}\log\sqrt{\alpha_t}}{\mathrm{d}t}\sigma_t^2$. The reverse process enables generation as it converts a Gaussian noise into the image.

### 2.2 Diffusion Guidance

In diffusion control, we aim to sample $\boldsymbol{x}_0$ given a condition $\boldsymbol{y}$. The conditional score function is expressed as follows:

$$\nabla_{\boldsymbol{x}_t} \log p_t(\boldsymbol{x}_t|\boldsymbol{y}) = \nabla_{\boldsymbol{x}_t} \log p_t(\boldsymbol{x}_t) + \nabla_{\boldsymbol{x}_t} \log p_t(\boldsymbol{y}|\boldsymbol{x}_t). \tag{2}$$

---

[2]The code is available at
https://github.com/BIGKnight/Understanding-Training-free-Diffusion-Guidance

The conditions are specified by the output of a neural network, and the energy is quantified by the corresponding loss function. If $\ell(f_\phi(\cdot), \cdot)$ represents the loss function as computed by neural networks, then the distribution of the clean data is expected to follow the following formula [2, 49, 37]:

$$p_0(\boldsymbol{x}_0|\boldsymbol{y}) \propto p_0(\boldsymbol{x}_0)\exp(-\ell(f_\phi(\boldsymbol{x}_0), \boldsymbol{y})). \tag{3}$$

For instance, consider a scenario where the condition is the object location. In this case, $f_\phi$ represents a fastRCNN architecture, and $\ell$ denotes the classification loss and bounding box loss. By following the computations outlined in [27], we can derive the exact formula for the second term in the RHS of (2) as:

$$\nabla_{\boldsymbol{x}_t} \log p_t(\boldsymbol{y}|\boldsymbol{x}_t) = \nabla_{\boldsymbol{x}_t} \log \mathbb{E}_{p(\boldsymbol{x}_0|\boldsymbol{x}_t)}[\exp(-\ell(f_\phi(\boldsymbol{x}_0), \boldsymbol{y})]. \tag{4}$$

**Classifier guidance** [8] involves initially training a time-dependent classifier to predict the output of the clean image $\boldsymbol{x}_0$ based on noisy intermediate representations $\boldsymbol{x}_t$ during the diffusion process, i.e., to train a time-dependent classifier $f_\psi(\boldsymbol{x}_t, t)$ such that $f_\psi(\boldsymbol{x}_t, t) \approx f_\phi(\boldsymbol{x}_0)$. Then the gradient of the time-dependent classifier is used for guidance, given by $\nabla_{\boldsymbol{x}_t} \log p_t(\boldsymbol{y}|\boldsymbol{x}_t) := -\nabla_{\boldsymbol{x}_t}\ell(f_\psi(\boldsymbol{x}_t, t), \boldsymbol{y})$. This term equals (4) if the loss is cross-entropy.

**Training-free loss-based guidance** [2, 49, 6] puts the expectation in (4) inside the loss function:

$$\nabla_{\boldsymbol{x}_t} \log p_t(\boldsymbol{y}|\boldsymbol{x}_t) := \nabla_{\boldsymbol{x}_t} \log \left[\exp(-\ell(f_\phi(\mathbb{E}_{p(\boldsymbol{x}_0|\boldsymbol{x}_t)}(\boldsymbol{x}_0)), \boldsymbol{y})\right] \overset{(a)}{=} -\nabla_{\boldsymbol{x}_t}\ell\left[f_\phi\left(\frac{\boldsymbol{x}_t - \sigma_t\boldsymbol{\epsilon}_\theta(\boldsymbol{x}_t, t)}{\sqrt{\alpha_t}}\right), \boldsymbol{y}\right], \tag{5}$$

where (a) uses Tweedie's formula $\mathbb{E}_{p(\boldsymbol{x}_0|\boldsymbol{x}_t)}(\boldsymbol{x}_0) = \frac{\boldsymbol{x}_t - \sigma_t\boldsymbol{\epsilon}_\theta(\boldsymbol{x}_t, t)}{\sqrt{\alpha_t}}$. Leveraging this formula permits the use of a pretrained off-the-shelf network designed for processing clean data. The gradient of the last term in the energy function is obtained via backpropagation through both the guidance network and the diffusion backbone.

## 3   Analysis of Training-Free Guidance

### 3.1   How does Training-free Guidance Work?

**On the difficulty of approximating $\nabla_{\boldsymbol{x}_t} \log p_t(\boldsymbol{y}|\boldsymbol{x}_t)$ in high-dimensional space.** Despite being intuitive, [27] has shown that training-free guidance in (5) does not offer an approximation to the true energy in (4). The authors of [37] consider to directly approximate (4) with a Gaussian distribution:

$$\begin{aligned}
\nabla_{\boldsymbol{x}_t} \log \mathbb{E}_{p(\boldsymbol{x}_0|\boldsymbol{x}_t)}[\exp(-\ell(f_\phi(\boldsymbol{x}_0), \boldsymbol{y})] &\overset{(a)}{\approx} \nabla_{\boldsymbol{x}_t} \log \mathbb{E}_{q(\boldsymbol{x}_0|\boldsymbol{x}_t)}[\exp(-\ell(f_\phi(\boldsymbol{x}_0), \boldsymbol{y})] \\
&\approx \nabla_{\boldsymbol{x}_t} \log \frac{1}{n}\sum_{i=1}^n \exp(-\ell(f_\phi(\boldsymbol{x}_0^i), \boldsymbol{y})), \quad \boldsymbol{x}_0^i \sim q(\boldsymbol{x}_0|\boldsymbol{x}_t),
\end{aligned} \tag{6}$$

where $q(\boldsymbol{x}_0|\boldsymbol{x}_t)$ is chosen as $\mathcal{N}(\mathbb{E}_{p(\boldsymbol{x}_0|\boldsymbol{x}_t)}(\boldsymbol{x}_0), r_t^2\boldsymbol{I})$ and $r_t$ is a tunable parameter. As demonstrated in [37], the approximation is effective for one-dimensional distribution. However, we find that the approximation denoted by (a) does not extend to high-dimensional data (e.g., images) if the surrogate distribution $q$ is sub-Gaussian. This is due to the well-known high-dimensional probability phenomenon [43] that if $q$ has sub-Gaussian coordinates (e.g., iid and bounded), then $q(\boldsymbol{x}_0|\boldsymbol{x}_t)$ tends to concentrate on a spherical shell centered at $\mathbb{E}_{p(\boldsymbol{x}_0|\boldsymbol{x}_t)}(\boldsymbol{x}_0)$ with radius $r_t$ (details are in Appendix C.1). Since the spherical shell represents a low-dimensional manifold with zero measure in the high-dimensional space, there is a significant likelihood that the supports of $p(\boldsymbol{x}_0|\boldsymbol{x}_t)$ and $q(\boldsymbol{x}_0|\boldsymbol{x}_t)$ do not overlap, rendering the approximation (a) ineffective.

**Understanding training-free guidance from an optimization perspective.** We instead analyze the training-free guidance from the optimization perspective. Intuitively, in each step, the gradient is taken and the loss of the guidance network decreases. At the initial stage of the diffusion ($t$ is large), the diffusion trajectory can exhibit substantial deviations between adjacent steps and may increase the objective value. So the objective value will oscillate at the beginning. When $t$ is smaller, the change to the sample is more fine-grained, leading to a bounded change in the objective value. Therefore, the objective value is guaranteed to decrease when $t$ is small, as showing in the next proposition.

**Proposition 3.1.** *Assume that the guidance loss function $\ell(f_\phi(\boldsymbol{x}_0), \boldsymbol{y})$ is $\mu$-PL (defined in Defintion D.2 in appendix) and $L_f$-Lipschitz with respect to clean images $\boldsymbol{x}_0$, and the score function*

$\nabla \log p_t(\boldsymbol{x}_t)$ is $L_p$-Lipschitz (defined in Defintion D.1 in appendix) with respect to noisy image $\boldsymbol{x}_t$. Denote $\lambda_{\min}$ as the minimum eigenvalue of the (semi)-definite matrix $Cov[\boldsymbol{x}_0|\boldsymbol{x}_t]$. Then the following conditions hold: (1) Consider the loss function $\ell_t(\boldsymbol{x}_t) = \ell\left(f_\phi\left(\frac{\boldsymbol{x}_t + \sigma_t^2 \nabla \log p_t(\boldsymbol{x}_t)}{\sqrt{\alpha_t}}\right), \boldsymbol{y}\right)$ and denote $\kappa_1 = \frac{\mu \lambda_{\min}^2}{L_f(1+L_p)\sqrt{\alpha_t}\sigma_t^4}$. After one gradient step $\hat{\boldsymbol{x}}_t = \boldsymbol{x}_t - \eta \nabla_{\boldsymbol{x}_t} \ell_t(\boldsymbol{x}_t), \eta = \frac{\sqrt{\alpha_t}}{L_f(1+L_p)}$, we have $\ell_t(\hat{\boldsymbol{x}}_t) \le (1 - \kappa_1)\ell_t(\boldsymbol{x}_t)$; (2) Consider a diffusion process that adheres to a bounded change in the objective function such that for any diffusion step, i.e., $\ell_{t-1}(\boldsymbol{x}_{t-1}) \le \frac{\ell_t(\hat{\boldsymbol{x}}_t)}{(1-\kappa_2)}$ for some $\kappa_2 < \kappa_1$, then the objective function converges at a linear rate, i.e., $\ell_{t-1}(\boldsymbol{x}_{t-1}) \le \frac{1-\kappa_1}{1-\kappa_2}\ell_t(\boldsymbol{x}_t)$.

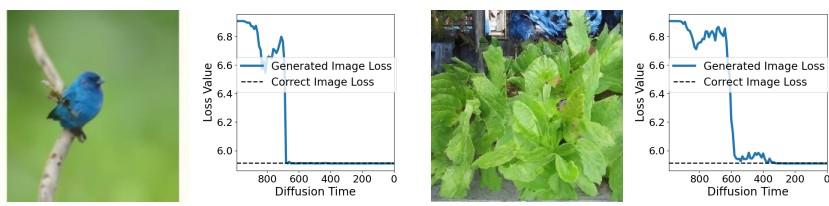

(a) A successful guidance.      (b) A failure guidance.

Figure 1: The classifier loss of a successful and a failure guidance example. The target class is "indigo bird".

The proof is given in Appendix D.2. The Lipschitz continuity and PL conditions are basic assumptions in optimization, and it has been shown that neural networks can locally satisfy these conditions [5]. These assumptions, while not essential, simplify the proof and results for clarity, similar to the assumptions taken in [26]. The optimization perspective clarifies the mystery of why the guidance weights (i.e., the step size in optimization) should be carefully selected with respect to the guidance function and time $t$. For example, in [49], most guidance weights $\eta$ are chosen to be proportional to $\sqrt{\alpha_t}$ and dependent on guidance network, which differs from the weights used in classifier guidance and aligns with our step size in Proposition 3.1.

Then we empirically verify Proposition 3.1 via experiments in Figure 1. We use ResNet-50 trained on clean images to guide ImageNet pretrained diffusion models. The loss value at each diffusion step is plotted. As a reference, we choose 100 images from the class "indigo bird" in ImageNet training set and compute the loss value, which is referred to as "Correct Image Loss" in the figure. The objective value oscillates when $t$ is large, followed by a swift decrease, which verifies our analysis. More convergence figures are given in Figure 7 in Appendix.

An intriguing aspect of the theory is that the loss remains low regardless of the success of the guidance, akin to the loss associated with correct images. Figure 1b demonstrates this phenomenon: despite the absence of an indigo bird in the image, the loss is still minimal. This phenomenon can be attributed to the effect of misaligned gradients, which is explored in detail in the following subsection.

## 3.2 Limitations of Training-free Guidance

In this subsection, we examine the disadvantages of employing training-free guidance networks as opposed to training-based classifier guidance.

**Training-free guidance is more sensitive to the misaligned gradient.** Adversarial gradient is a significant challenge for neural networks, which refers to minimal perturbations deliberately applied to inputs that can induce disproportionate alterations in the model's output [38]. The resilience of a model to adversarial gradients is often analyzed through the lens of its Lipschitz constant [34]. If the model has a lower Lipschitz constant, then the output is less sensitive to the input perturbations and thus is more robust.

In the classifier or training-free guidance, the gradient of the guidance network is added to the image. In contrast to yielding a direction that meaningfully minimizes the loss, the adversarial gradient primarily serves to minimize the loss in a manner that is not necessarily aligned with the intended guidance direction. As a result, we refer the adversarial gradient of guidance network as the *misaligned gradient* in diffusion guidance.

Compared with the off-the-shelve guidance network used in training-free guidance, time-dependent classifiers are trained on noise-augmented images. Our finding is that adding Gaussian noise improves the Lipschitzness of the guidance network. This transition mitigates the misaligned gradient challenge by inherently enhancing the model's robustness to such perturbations, as shown in the next proposition.

**Proposition 3.2.** *(Time-dependent network is more robust and smooth) Given a bounded loss function* $\ell(\boldsymbol{x}) \leq C$, *the loss* $\hat{\ell}(\boldsymbol{x}) = \mathbb{E}_{\boldsymbol{\epsilon} \sim \mathcal{N}(0, \boldsymbol{I})}[\ell(\boldsymbol{x} + \sigma_t \boldsymbol{\epsilon})]$ *is* $C \sqrt{\frac{2}{\pi \sigma_t^2}}$*-Lipschitz and* $\nabla \hat{\ell}$ *is* $\frac{2C}{\sigma_t}$*-Lipschitz.*

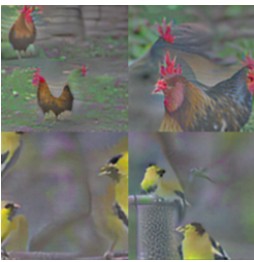 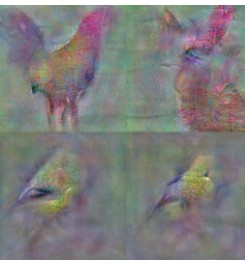 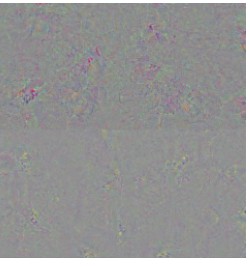 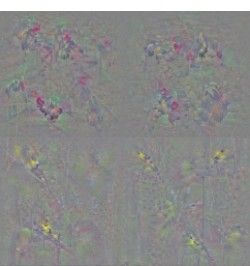

(a) Adversarially robust classifier. (b) Time-dependent classifier. (c) Off-the-shelf ResNet-50 classifier. (d) ResNet-50 with random augmentation.

Figure 2: Gradients of different classifiers on random backgrounds. The images in the first row correspond to the target class "cock", and the second row to "goldfinch".

The proof is given in Appendix D.3. We then support Proposition 3.2 with both qualitative and quantitative experiments. For qualitative experiments, we present visualizations of the accumulated gradients for both the time-dependent and off-the-shelf time-independent classifiers corresponding to different classes in Figure 2b and Figure 2c, respectively. These visualizations are generated by initializing an image with a random background and computing 1000 gradient steps for each classifier. For the time-dependent classifier, the input time for the $t$-th gradient step is $1000 - t$. The images are generated purely by the classifier gradients without diffusion involved. For comparative analysis, we include the accumulated gradient of an adversarially robust classifier [35], as shown in Figure 2a, which has been specifically trained to resist misaligned (adversarial) gradients. The resulting plots reveal a stark contrast: the gradient of the time-dependent classifier visually resembles the target image, whereas the gradient of the time-independent classifier does not exhibit such recognizability. This observation suggests that off-the-shelf time-independent classifiers are prone to generating misaligned gradients for guidance compared to the time-dependent classifier used in classifier guidance. The quantitative experiments are given in Table 4 in Appendix.

Figure 2 provides a more intuitive visual explanation of diffusion guidance compared to the existing formula-based approaches as shown in (7). The gradient produced by the guidance network represents a valid image. When these gradients are incorporated into the images, the diffusion model is able to identify the object and enhance it into a clearer and more vivid representation.

**Training-free guidance slows down the convergence of reverse ODE.** The efficiency of an algorithm in solving reverse ordinary differential equations is often gauged by the number of non-linear function estimations (NFEs) required to achieve convergence. This metric is vital for algorithmic design, as it directly relates to computational cost and time efficiency [36]. In light of this, we explore the convergence rates associated with various guidance paradigms, beginning our analysis with a reverse ODE framework that incorporates a generic gradient guidance term. The formula is expressed as

$$\frac{\mathrm{d}\boldsymbol{x}_t}{\mathrm{d}t} = f(t)\boldsymbol{x}_t + \frac{g^2(t)}{2}(\boldsymbol{\epsilon}_\theta(\boldsymbol{x}_t, t) + \nabla_{\boldsymbol{x}_t} v(\boldsymbol{x}_t, t)), \tag{7}$$

where $h(\cdot, \cdot)$ can be either a time-dependent classifier or a time-independent classifier with Tweedie's formula. The subsequent proposition elucidates the relationship between the discretization error and the smoothness of the guidance function.

**Proposition 3.3.** *Let* $u(\boldsymbol{x}_t, t) = \boldsymbol{\epsilon}_\theta(\boldsymbol{x}_t, t) + \nabla_{\boldsymbol{x}_t} v(\boldsymbol{x}_t, t)$ *in* (7), $h_{\max} = \max_t \frac{1}{2}\left[\log(\frac{\alpha_t}{1-\alpha_t}) - \log(\frac{\alpha_{t-1}}{1-\alpha_{t-1}})\right]$. *Assume we run DDIM solver for* $M$ *steps and* $M = O(1/h_{\max})$. *Then the error is bounded by* $O((1 + L^M)/M)$.

The proof is given in Appendix D.4. Proposition 3.2 establishes that time-dependent classifiers exhibit superior gradient Lipschitz constants compared to their off-the-shelf time-independent counterparts. This disparity in smoothness slows down the convergence for training-free guidance methods, necessitating a greater number of NFEs to achieve the desired level of accuracy when compared to classifier guidance. To provide quantitative support, we compare the convergence speed of training-based PPAP [12] and training-free FreeDoM in Table 5 in Appendix.

## 4 Improving Training-free Guidance

In this section, we propose to adopt random augmentation to mitigate the misaligned gradient issue, and Polyak step size [15] to mitigate the convergence issue. In addition to these two techniques, our method and baselines will also incorporate a trick named time travel, often referred to as "restart sampling", and its theoretical framework is detailed in [46].

---

**Algorithm 1** Random Augmentation

**for** $t = T, \cdots, 0$ **do**
$\quad x_{t-1} = \text{DDIM}(x_t)$
$\quad \hat{x}_0 = \frac{x_t - \sigma_t \epsilon_\theta(x_t, t)}{\sqrt{\alpha_t}}$ $\quad \triangleright$ Tweedie's formula
$\quad g_t = \frac{1}{|\mathcal{T}|} \sum_{T \in \mathcal{T}} \nabla_{x_t} \ell(f_\phi(T(\hat{x}_0)), y)$
$\quad x_{t-1} = x_{t-1} - \eta \cdot g_t$
**end for**

---

**Algorithm 2** Polyak Step Size

**for** $t = T, \cdots, 0$ **do**
$\quad x_{t-1} = \text{DDIM}(x_t)$
$\quad \hat{x}_0 = \frac{x_t - \sigma_t \epsilon_\theta(x_t, t)}{\sqrt{\alpha_t}}$ $\quad \triangleright$ Tweedie's formula
$\quad g_t = \nabla_{x_t} \ell(f_\phi(\hat{x}_0), y)$
$\quad x_{t-1} = x_{t-1} - \eta \cdot \frac{\|\epsilon_\theta(x_t, t)\|}{\|g_t\|_2^2} \cdot g_t$
**end for**

---

### 4.1 Random Augmentation

As established by Proposition 3.2, the introduction of Gaussian perturbations enhances the Lipschitz property of a neural network. A direct application of this principle involves creating multiple noisy instances of an estimated clean image and passing them into the guidance network, a method analogous to the one described in (6). However, given the high-dimensional nature of image data, achieving a satisfactory approximation of the expected value necessitates an impractically large number of noisy copies. To circumvent this issue, we propose an alternative strategy that employs a diverse set of data augmentations in place of solely adding Gaussian noise. This approach effectively introduces perturbations within a lower-dimensional latent space, thus requiring fewer samples. The suite of data augmentations utilized, denoted by $\mathcal{T}$, is derived from the differentiable data augmentation techniques outlined in [51], which encompasses transformations such as translation, resizing, color adjustments, and cutout operations. The details are shown in Algorithm 1 and the rationale is shown in the following proposition.

**Proposition 4.1.** *(Random augmentation improves smoothness) Given a bounded non-Lipschitz loss function $\ell(x)$, the loss $\hat{\ell}(x) = \mathbb{E}_{\epsilon \sim p(\epsilon)}[\ell(x + \epsilon)]$ is $C \int_{\mathbb{R}^n} \|\nabla p(t)\|_2 \mathrm{d}t$-Lipschitz and its gradient is $C \int_{\mathbb{R}^n} \|\nabla^2 p(t)\|_{op} \mathrm{d}t$-Lipschitz.*

The proof is shown in Appendix D.5. Echoing the experimental methodology delineated in Section 3.2, we present an analysis of the accumulated gradient effects when applying random augmentation to a ResNet-50 model. Specifically, we utilize a set of $|\mathcal{T}| = 10$ diverse transformations as our augmentation strategy. The results of this experiment are visualized in Figure 2d, where the target object's color and shape emerge in the gradient profile. This observation suggests that the implementation of random augmentation can alleviate the misaligned gradient issue. The quantitative effect of random augmentation is given in Table 4 in Appendix. The computational efficiency of random augmentation is further discussed in Appendix C.4.

### 4.2 Polyak Step Size

In Section 3.1, we analyzed training-free guidance from the optimization perspective. To accelerate the convergence, gradient step size should be adaptive to the gradient landscape. We adopt Polyak step size, which has near-optimal convergence rates under various conditions [15]. The algorithm is

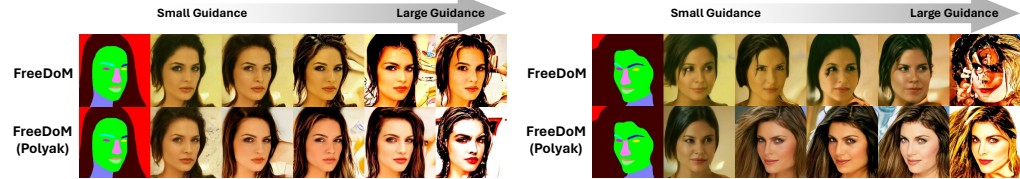

(a) When the initialization is proximate to the specified conditions, both step sizes perform satisfactorily.

(b) When the initialization deviates from the conditions, only the Polyak step size guides effectively.

Figure 3: The effects of step size.

shown in Algorithm 2 and the term $\|\boldsymbol{\epsilon}_\theta(\boldsymbol{x}_t, t)\|$ is used to both estimate the gap to optimal values and balance the magnitude of diffusion term and guidance term.

We implement Polyak step size within the context of a training-free guidance framework called FreeDoM [49] and benchmark the performance of this implementation using the DDIM sampler with 50 steps. As shown in Figure 3, FreeDoM is unable to effectively guide the generation process when faced with a significant discrepancy between the unconditional generation and the specified condition. An illustrative example is the difficulty in guiding the model to generate faces oriented to the left when the unconditionally generated faces predominantly orient to the right, as shown in Figure 3b. This challenge, which arises due to the insufficiency of 50 steps for convergence under the condition, is ameliorated by substituting gradient descent with adaptive step size, thereby illustrating the benefits of employing a better step size in the guidance process. The quantitative experiments are given in Table 5 in Appendix.

## 5 Experiments

In this section, we evaluate the efficacy of our proposed techniques across various diffusion models and guidance conditions. We compare our methods with established baselines: Universal Guidance (UG) [2], Loss-Guided Diffusion with Monte Carlo (LGD-MC) [37], Training-Free Energy-Guided Diffusion Models (FreeDoM) [49], and Manifold Preserving Guided Diffusion (MPGD) [16]. LGD-MC utilizes (6) while UG and FreeDoM are built on (5). MPGD utilizes an auto-encoder to ensure the manifold constraints. Furthermore, time travel trick (Algorithm 3) is adopted in UG, FreeDoM, and MPGD to improve the performance. Please refer to Appendix B for details of baselines. For the sampling method, DDIM with 100 steps is adopted as in [49, 37]. The method "Ours" is built on FreeDoM, with Polyak step size and random augmentation.

### 5.1 Guidance to CelebA-HQ Diffusion

In this subsection, we adopt the experimental setup from [49]. Specifically, we utilize the CelebA-HQ diffusion model [19] to generate high-quality facial images. We explore three guidance conditions: segmentation, sketch, and text. For segmentation guidance, BiSeNet [48] generates the facial segmentation maps, with an $\ell_2$-loss applied between the estimated map of the synthesized image and the provided map. Sketch guidance involves using the method from [45] to produce facial sketches, where the loss function is the $\ell_2$-loss between the estimated sketch of $\hat{\boldsymbol{x}}_0$ and the given sketch. For text guidance, we employ CLIP [32] as both the image and text encoders, setting the loss to be the $\ell_2$ distance between the image and text embeddings.

We randomly select 1000 samples each of segmentation maps, sketches, and text descriptions. The comparative results are presented in Table 1. Consistent with [49], the time-travel number for all methods is set to $s = 1$. Figure 4 displays a random selection of the generated images. More image samples are provided in the supplementary materials. We find that the baselines failed to guide if the condition differs from unconditionally generated images significantly, as discussed in Section 4.2.

### 5.2 Guidance to ImageNet Diffusion

For the unconditional ImageNet diffusion, we employ text guidance in line with the approach used in FreeDoM and UG [2, 49]. We utilize CLIP-B/16 as the image and text encoder, with cosine similarity

| Methods | Segmentation maps | | Sketches | | Texts | |
|---|---|---|---|---|---|---|
| | Distance↓ | FID↓ | Distance↓ | FID↓ | Distance↓ | FID↓ |
| UG [2] | 2247.2 | 39.91 | 52.15 | 47.20 | 12.08 | 44.27 |
| LGD-MC [37] | 2088.5 | 38.99 | 49.46 | 54.47 | 11.84 | 41.74 |
| FreeDoM [49] | 1657.0 | 38.65 | 34.21 | 52.18 | 11.17 | 46.13 |
| MPGD-Z [16] | 1976.0 | 39.81 | 37.23 | 54.18 | 10.78 | 42.45 |
| Ours | **1575.7** | **33.31** | **30.41** | **41.26** | **10.72** | **41.25** |

Table 1: The performance comparison of various methods on CelebA-HQ with different types of zero-shot guidance. The experimental settings adhere to Table 1 of [49].

| Methods | LGD-MC [37] | UG [2] | FreeDoM [49] | MPGD-Z [16] | Ours |
|---|---|---|---|---|---|
| CLIP Score↑ | 24.3 | 25.7 | 25.9 | 25.1 | **27.7** |

Table 2: The performance comparison of various methods on unconditional ImageNet with zero-shot text guidance. We compare various methods using ImageNet pretrained diffusion models with CLIP-B/16 guidance. For evaluating performance, the CLIP score is computed using CLIP-L/14.

serving as the loss function to measure the congruence between the image and text embeddings. To evaluate performance and mitigate the potential for high-scoring adversarial images, we use CLIP-L/14 for computing the CLIP score. In FreeDoM and MPGD-Z, resampling is conducted for time steps ranging from 800 to 300, with the time-travel number fixed at 10, as described in [49]. Given that UG resamples at every step, we adjust its time-travel number $s = 5$ to align the execution time with that of FreeDoM. The textual prompts for our experiments are sourced from [25]. The comparison of different methods is depicted in Table 2. The corresponding randomly selected images are illustrated in Figure 5. The table indicates that our method achieves the highest consistency with the provided prompts. As shown in Figure 5, LGD-MC and MPGD tend to overlook elements of the prompts. Both UG and FreeDoM occasionally produce poorly shaped objects, likely influenced by misaligned gradients. Our approach addresses this issue through the implementation of random augmentation. Additionally, none of the methods successfully generate images that accurately adhere to positional prompts such as "left to" or "below". This limitation is inherent to CLIP and extends to all text-to-image generative models [41]. More image samples are provided in the supplementary materials.

### 5.3 Guidance to Human Motion Diffusion

In this subsection, we extend our evaluation to human motion generation using the Motion Diffusion Model (MDM) [40], which represents motion through a sequence of joint coordinates and is trained on a large corpus of text-motion pairs with classifier-free guidance. We apply the targeting guidance and object avoidance guidance as described in [37]. Let $x_0(t)$ denote the joint coordinates at time $t$, $y_t$ the target location, $y_{obs}$ the obstacle location, $r$ the radius of the objects, and $T$ the total number of frames. The loss function is defined as follows:

$$\ell = \|y_t - x_0(T)\|_2^2 + \sum_i \text{sigmoid}(-(\|x_0(i) - y_{obs}\| - r) \times 50) \times 100. \tag{8}$$

Our experimental configuration adheres to the guidelines set forth in [37]. We assess the methods using the targeting loss (the first term in (8)), the object avoidance loss (the second term in (8)), and the CLIP score calculated by MotionCLIP [39]. In this application, MPGD-Z cannot be applied as there are no auto-encoder. MPGD w/o proj suffers from the shortcut and cannot achieve good performance, as discussed in Appendix B.2. In our method, random augmentation is omitted because the guidance is not computed by neural networks so the adversarial gradient issues are not obvious. The quantitative results of our investigation are summarized in Table 3, while Figure 6 showcases randomly selected samples. Our methods exhibit enhanced control quality over the generated motion. The videos are provided in the supplementary materials.

| Methods | "Backwards" | | "Balanced Beam" | | "Walking" | | "Jogging" | |
|---|---|---|---|---|---|---|---|---|
| | Loss↓ | CLIP↑ | Loss↓ | CLIP↑ | Loss↓ | CLIP↑ | Loss↓ | CLIP↑ |
| Unconditional [40] | $3.55 + 9.66$ | 65.6 | $47.92 + 0$ | **70.8** | $48.88 + 0$ | 37.6 | $144.84 + 0$ | **61.72** |
| FreeDoM [49] | $1.09 + 6.63$ | 67.23 | $9.83 + 4.48$ | 62.65 | $1.64 + 7.55$ | 40.12 | $34.95 + 7.83$ | 58.74 |
| LGD-MC [37] | $0.98 + 6.48$ | 67.31 | $4.42 + 0.02$ | 63.13 | $1.30 + 0.39$ | 38.82 | $6.12 + 2.38$ | 57.89 |
| Ours | **0.68+1.32** | **67.50** | **1.13+0.30** | 63.02 | **0.43+0.31** | **40.40** | **2.93+1.15** | 60.03 |

Table 3: Comparison of various methods on MDM with zero-shot targeting and object avoidance guidance. Loss is reported as a two-component metric: the first part is the MSE between the target and the actual final position of the individual; the second part measures the object avoidance loss.

## 5.4 Related Work on Training-Free Guidance

Due to space limitation, we only introduce the related work on training-free guidance while leaving more related work in Appendix A. The current training-free guidance strategies for diffusion models can be divided into two primary categories. The first category is the loss-based guidance in this paper, which is universally applicable to universal control formats and diffusion models. These methods predict a clean image, subsequently leveraging pretrained networks to guide the diffusion process. Central to this approach are the algorithms based on (5), which have been augmented through techniques like time-travel [2, 49] and the introduction of Gaussian noise [37]. The adjoint sensitivity method [31] and spherical Gaussian constraint [47] have been adopted to estimate a more accurate guidance gradient. Extensions of these algorithms have found utility in domains with constrained data-condition pairs, such as molecule generation [14], and in scenarios necessitating zero-shot guidance, like open-ended goals in offline reinforcement learning [44]. In molecular generation and offline reinforcement learning, they outperform training-based alternatives as additional training presents challenges. This paper delves deeper into the mechanics of this paradigm and introduces a suite of enhancements to bolster its performance. The efficacy of our proposed modifications is demonstrated across image and motion generation, with promising potential for generalization to molecular modeling and reinforcement learning tasks.

The second category of training-free guidance is tailored to text-to-image or text-to-video diffusion models, which is based on insights into their internal backbone architecture. For instance, object layout and shape have been linked to the cross-attention mechanisms [17], while network activations have been shown to preserve object appearance [42]. These understandings facilitate targeted editing of object layout and appearance (Diffusion Self-Guidance [11]) and enable the imposition of conditions in ControlNet through training-free means (FreeControl [30]). Analyzing these methodologies is challenging due to their reliance on emergent representations during training. Nonetheless, certain principles from this paper remain relevant; for example, as noted in Proposition 3.3, these methods often necessitate extensive diffusion steps, with instances such as [30, 11] employing 1000 steps. A thorough examination and refinement of these techniques remain an avenue for future research.

## 6 Conclusions

In this paper, we conducted a comprehensive investigation into training-free guidance, which employs pretrained diffusion models and guides them using the off-the-shelf trained on clean images. Our exploration delved into the underlying mechanisms and fundamental limits of these models. Moreover, we proposed a set of enhancement techniques and verified their effectiveness both theoretically and empirically.

**Limitations.** Despite our efforts to mitigate the shortcomings of training-free methods and enhance their performance, certain limitations remain. Notably, the refined training-free guidance still necessitates a higher number of NFEs when compared with extensive training methods such as classifier-free guidance. This is because misaligned gradient cannot be fully eliminated without training.

**Ethical Consideration.** Similar to other models designed for image creation, our model also has the unfortunate potential to be used for creating deceitful or damaging material. We pledge to restrict the usage of our model exclusively to the realm of research to prevent such misuse.

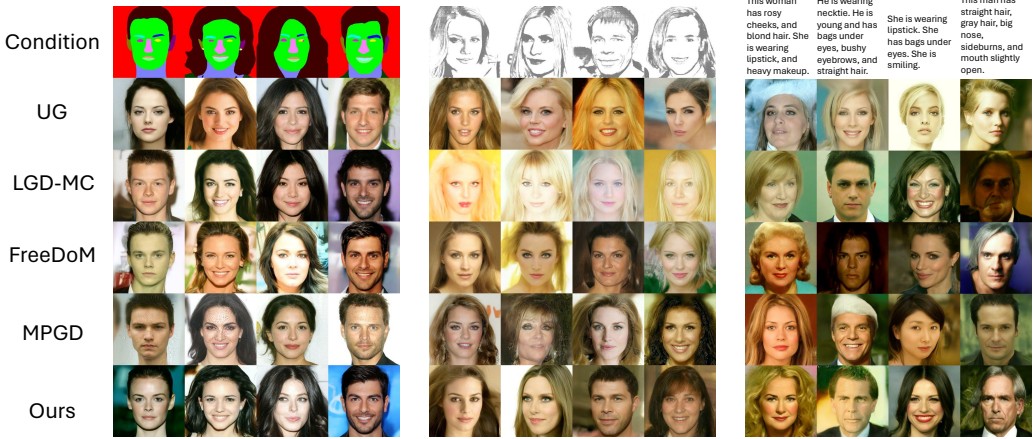

Figure 4: Qualitative results of CelebA-HQ with zero-shot segmentation, sketch, and text guidance. The images are randomly selected.

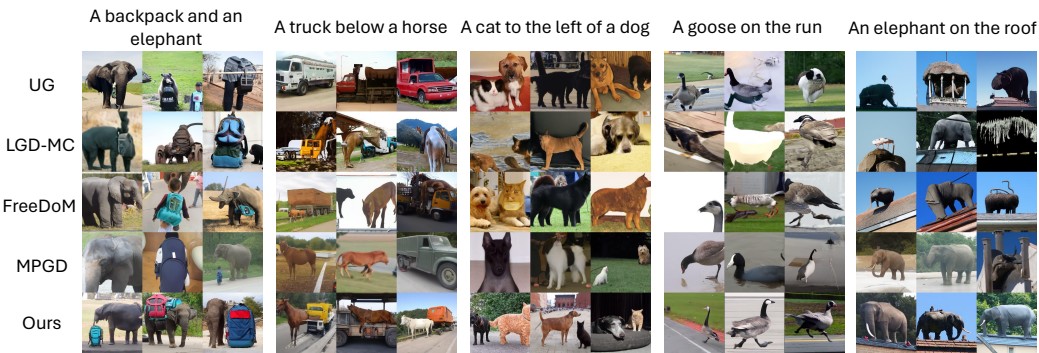

Figure 5: Qualitative results of ImageNet model with zero-shot text guidance. The images are randomly selected.

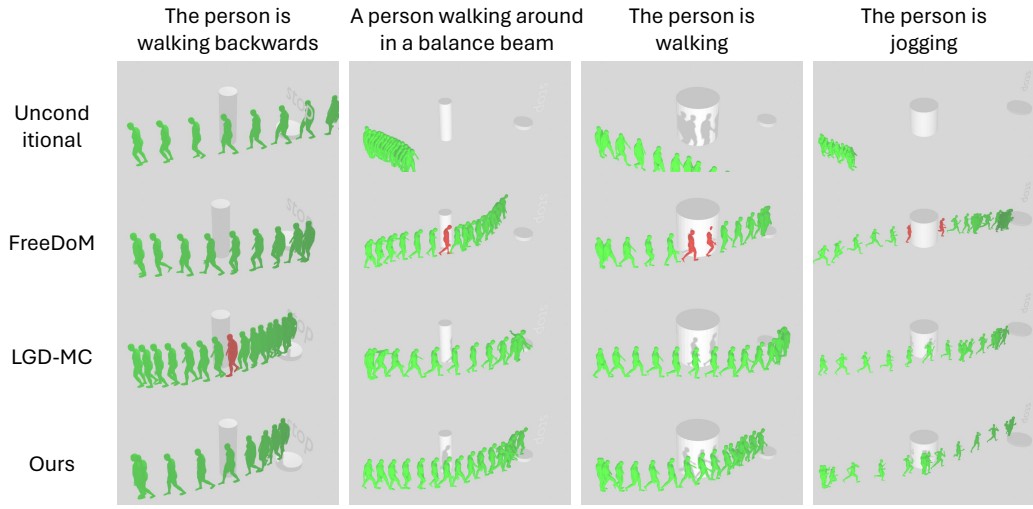

Figure 6: Qualitative results of human motion diffusion with zero-shot object avoidance and targeting guidance. Instances of intersection with obstacles are highlighted by marking the person in red. The trajectories are randomly selected.

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

# A  Related Works

## A.1  Training-based Gradient Guidance

The training-based gradient guidance paradigm, such as classifier guidance, is a predominant approach for diffusion guidance. The core objective is to train a time-dependent network that approximates $p_t(\boldsymbol{y}|\boldsymbol{x}_t)$ in the RHS of (2), and to utilize the resulting gradient as guidance. The most well-known example is classifier guidance, which involves training a classifier on noisy images. However, classifier guidance is limited to class conditions and is not adaptable to other forms of control, such as image and text guidance. To address this limitation, there are two main paradigms. The first involves training a time-dependent network that aligns features extracted from both clean and noisy images, as described by [24]. The training process is outlined as follows:

$$\min_{\psi} \quad \mathbb{E}_{p(\boldsymbol{x}_0, \boldsymbol{x}_t)} d(f_\psi(\boldsymbol{x}_t, t), f_\phi(\boldsymbol{x}_0)),$$

where $d(\cdot, \cdot)$ represents a loss function, such as cross-entropy or the $\ell_2$ norm. If time-dependent networks for clean images are already available, training can proceed in a self-supervised fashion without the need for labeled data. The second paradigm, as outlined by [21], involves training an energy-based model to approximate $p_t(\boldsymbol{y}|\boldsymbol{x}_t)$. The training process is described as follows:

$$\min_{\psi} \quad \mathbb{E}_{p(\boldsymbol{x}_0, \boldsymbol{x}_t)} |\ell(f_\psi(\boldsymbol{x}_t, t), \boldsymbol{y}) - \ell(f_\phi(\boldsymbol{x}_0), \boldsymbol{y})|.$$

However, it is observed in [27] that none of these methods can accurately approximate the true energy in (4). The authors of [27] propose an algorithm to learn the true energy. The loss function is a contrastive loss

$$\min_{\psi} \quad \mathbb{E}_{p(\boldsymbol{x}_0^i, \boldsymbol{x}_t^i)} \exp(-\ell(f_\phi(\boldsymbol{x}_0), \boldsymbol{y})) \left[ -\sum_{i=1}^{K} \log \frac{\exp(\ell(f_\psi(\boldsymbol{x}_t^i, t), \boldsymbol{y}^i)}{\sum_{j=1}^{K} \exp(-\ell(f_\psi(\boldsymbol{x}_t^i, t), \boldsymbol{y}^i))} \right],$$

where $(\boldsymbol{x}_0^i, \boldsymbol{x}_t^i)$ are $K$ paired data samples from $p(\boldsymbol{x}_0^i, \boldsymbol{y}^i)$. Theorem 3.2 in [27] proves that the optimal $f_{\psi^*}$ satisfied that $\nabla_{\boldsymbol{x}_t} \ell(f_{\psi^*}(\boldsymbol{x}_t^i, t), \boldsymbol{y}^i) = \nabla_{\boldsymbol{x}_t} p_t(\boldsymbol{y}|\boldsymbol{x}_t)$.

Although this paper focuses on training-free guidance, the findings in this paper can be naturally extended to all training-based gradient guidance schemes. Firstly, the issue of adversarial gradients cannot be resolved without additional training; hence, all the aforementioned methods are subject to adversarial gradients. Empirical evidence for this is presented in Fig. 2, which illustrates that the gradients from an adversarially robust classifier are markedly more vivid than those from time-dependent classifiers. Consequently, it is anticipated that incorporating additional adversarial training into these methods would enhance the quality of the generated samples. Secondly, since these methods are dependent on gradients, employing a more sophisticated gradient solver could further improve their NFEs.

## A.2  Adversarial Attack and Robustness

Adversarial attacks and robustness constitute a fundamental topic in deep learning [4]. An adversarial attack introduces minimal, yet strategically calculated, changes to the original data that are often imperceptible to humans, leading models to make incorrect predictions. The most common attacks are gradient-based, for example, the Fast Gradient Sign Method (FGSM) [13], Projected Gradient Descent (PGD) [29], Smoothed Gradient Attacks [1], and Momentum-Based Attacks [9]. An attack is akin to classifier guidance or training-free guidance, which uses the gradient of a pre-trained network for guidance. Should the gradient be adversarial, the guidance will be compromised. This paper establishes the relationship between training-free loss-guided diffusion models and adversarial attacks in two ways. Firstly, we prove that training-free guidance is more sensitive to an adversarial gradient. Secondly, in Section 4.2, we demonstrate that borrowing an adaptive gradient scheduler can improve convergence. The optimizers from adversarial attack literature may also expedite the convergence of the diffusion ODE.

# B  Baselines and Experimental Settings

## B.1  Details of the Baselines

We use the following training-free diffusion guidance methods as baselines for comparison:

- **Universal Guidance (UG) [2]** employs guidance as delineated in (5) and uses time-travel strategies outlined in Algorithm 3 to enhance performance. The time-travel trick is used for all time steps $t$. UG also utilizes backward guidance, which takes multiple gradient steps at each time step.

- **FreeDoM [49]** is also founded on (5) and time-travel trick. In addition, FreeDoM incorporates a time-dependent step size for each gradient guidance and judiciously selects the diffusion step for executing time-travel trick.

- **Loss-guided Diffusion with Monte Corlo (LGD-MC) [37]** utilizes guidance from (6) and we set $n = 10$ in the experiments.

- **Manifold Guided Preserving Diffusion (MPGD) [16]** takes the derivative with respect to estimated clean image $\mathbb{E}[\boldsymbol{x}_0|\boldsymbol{x}_t]$ instead of $\boldsymbol{x}_t$. Let $\boldsymbol{x}_{0|t} = \mathbb{E}[\boldsymbol{x}_0|\boldsymbol{x}_t]$, MPGD steps are expressed as following:

$$\nabla_{\boldsymbol{x}_t} \log p_t(\boldsymbol{y}|\boldsymbol{x}_t) := -\sqrt{\alpha_{t-1}} \nabla_{\boldsymbol{x}_{0|t}} \log \left[ \exp(-\ell(f_\phi(\boldsymbol{x}_{0|t}), \boldsymbol{y})) \right].$$

MPGD-Z adopts an additional auto-encoder to preserve manifold constraints. The details procedures of MPGD-Z are described in Algorithm 3 of [16].

## B.2 MPGD for Motion Diffusion

For the process of motion diffusion, the application of both MPGD-Z and MPGD-AE is precluded due to the absence of pretrained auto-encoders specific to motion diffusion. An implementation of MPGD without projection (MPGD w/o proj) was attempted for motion diffusion; however, it was unsuccessful in accurately navigating towards the target. This failure is attributed to the presence of spurious correlations within the targeting loss specific to MPGD, a phenomenon not observed in the other baseline methodologies. Sepcifically, the gradient formulation in MPGD is detailed as follows:

$$\text{grad\_MPGD} = \begin{cases} 2(\boldsymbol{y}_{\text{target}} - \boldsymbol{x}_{0|t}) & \text{if } t == T \\ 0 & \text{otherwise} \end{cases}$$

The gradient in FreeDoM and other methods is given by

$$\text{grad\_DPS} = 2(\boldsymbol{y}_{\text{target}} - \boldsymbol{x}_{0|t}) \cdot \frac{\boldsymbol{I} + \sigma_t^2 \nabla^2 \log p_t(\boldsymbol{x}_t)}{\sqrt{\alpha_t}}.$$

Analysis of the aforementioned equations reveals that, within the MPGD framework, only the final motion step is influenced by the gradient, a characteristic not shared by alternative methodologies. Consequently, this exclusive focus on the last step results in disproportionately strong guidance at this juncture, while earlier steps suffer from a lack of directional input. This imbalance may adversely affect the overall quality of the samples produced. Empirical observations substantiate that MPGD struggles to achieve targeted outcomes when a nuanced adjustment of step size is required. Given these limitations, MPGD has been excluded from the comparative analysis presented in Table 3.

## B.3 Prompts for Motion Diffusion

We follow the prompts and evaluation settings in [37]. The prompts are (i) "the person is walking backwards"; (ii) "a person walking around in a balance beam"; (iii) "the person is walking"; (iv) "the person is jogging". We consider three different directions for each prompt, and each direction has 10 random seeds, the metrics are then averaged together over the 30 synthesized motions.

## C More Discussions

### C.1 Concentration of Estimated Clean Samples

It has been demonstrated in [7] that, given a fixed $\boldsymbol{x}_0$, the distribution of the noisy data $\boldsymbol{x}_t$ is concentrated on a spherical shell. An extension of this theorem presented in the "High Dimensional Probability" textbook by Vershynin [43] elucidates that the conditional distribution $q(\boldsymbol{x}_0|\boldsymbol{x}_t)$ also exhibits concentration on a spherical shell, provided that its coordinates are sub-Gaussian.

**Theorem C.1.** *(Theorem 3.1.1 in [43]) Denote $\boldsymbol{x}_0 = \boldsymbol{x}_t + \boldsymbol{g} \in \mathbb{R}^n$, where $g = (g_1, \cdots, g_n)$. Assume that $g_i$ are independent identically distributed, $\mathbb{E}[g_i^2] = \frac{r_t^2}{n}$ and there exists constant $c_1$ such that $\mathbb{P}[g_i \geq t] \leq \exp(-c_1 t^2)$, then we have*

$$\mathbb{P}[|\|\boldsymbol{x}_t - \boldsymbol{x}_0\|_2^2 - r_t^2| \geq t] \leq \exp(-c_2 n t^2),$$

*where $c_2$ is a constant.*

Theorem C.1 establishes that the distribution $q(\boldsymbol{x}_0|\boldsymbol{x}_t)$ exhibits concentration on a spherical shell at an exponential rate. For high-dimensional data, such as images, it is reasonable to infer that $\boldsymbol{x}_0$ is predominantly situated on this spherical shell.

## C.2 Time-travel Trick

---
**Algorithm 3** Time Travel
---
 1: **for** $t = T, \cdots, 1$ **do**
 2:     **for** $i = 1, \cdots, s$ **do**
 3:         $\boldsymbol{x}_{t-1}^i = \text{DDIM with Guidance}(\boldsymbol{x}_t^{i-1})$
 4:         **if** $i < s$ **then**
 5:             $\beta_t = \alpha_t/\alpha_{t-1}, \boldsymbol{n} \sim \mathcal{N}(0, \boldsymbol{I})$
 6:             $\boldsymbol{x}_t^i = \sqrt{\beta_t}\boldsymbol{x}_{t-1}^i + \sqrt{1 - \beta_t}\boldsymbol{n}$
 7:         **end if**
 8:         $\boldsymbol{x}_{t-1}^0 = \boldsymbol{x}_{t-1}^s$
 9:     **end for**
10: **end for**

---

The technique of time-travel, also referred to as "resampling", has been proposed as a solution to complex generative problems [28], facilitating successful training-free guidance in tasks such as CLIP-guided ImageNet generation and layout guidance for stable diffusion, as illustrated in Figure 2 of [49] and Figure 8 of [2], respectively. The procedure of the time-travel trick is shown in Algorithm 3, which involves recursive execution of individual sampling steps.

## C.3 More Quantitative Experiments

**Two-stage convergence:** We plot the convergence curve of more images in Figure 5. The diffusion model is ImageNet diffusion and the guidance network is ResNet-50. The convergence is two-stage: the objective value oscillates when t is large, followed by a swift decrease, which verifies Proposition 3.1.

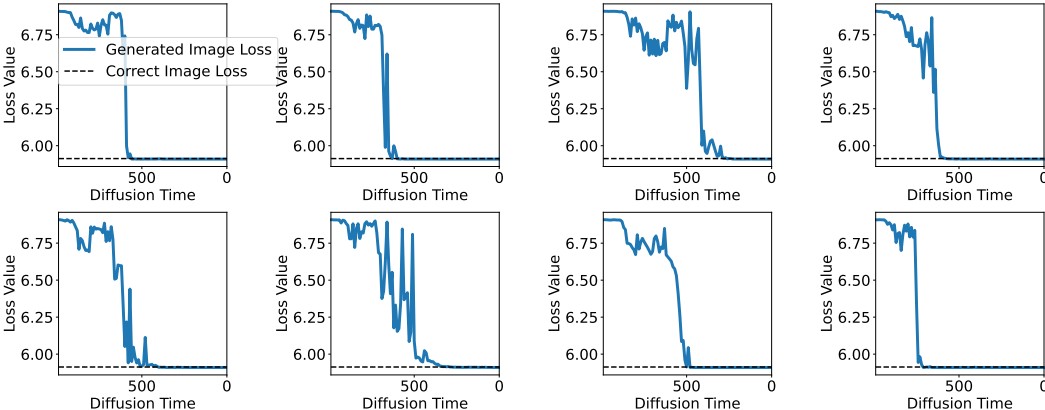

Figure 7: More experiments of the convergence plot.

**Misaligned gradients:** For quantitative experiments, we use robust ResNet-50 to assess the presence of misaligned gradient. Since ResNet-50 serves as our guidance network, a large loss gap between

the standard ResNet-50 and the robust variant indicates a severe adversarial gradient issue. In Table 4, the values represent the loss value. The columns denote the guidance network type, while the rows indicate the loss tested on either ResNet-50 or Robust ResNet-50. The values reported are the average over 1000 images, providing a quantitative evaluation. When using ResNet-50 for guidance, its loss on ResNet-50 is as low as that of real images from the same class (value 5.91). However, the loss for Robust ResNet-50 is significantly higher (value 6.17), suggesting susceptibility to misaligned gradients. By employing ResNet-50 with random augmentation for guidance, we observe a marked reduction in this gap (from 5.91 to 5.98), underscoring the effectiveness of our proposed method.

**Slower Convergence:** Table 5 aims to compare the convergence of training-free guidance with training-based guidance, specifically PPAP [12]. Given that classifier guidance presents a simple scenario and all methods converge relatively fast, it tends to obscure the observations. Consequently, we have directed our attention towards the more complex task of segmentation map guidance in Table 1. The columns in Table 5 list different methods, while the rows indicate the sampling steps. The values represent the "distance" mentioned in Table 1 and are averaged over 1000 images under the same conditions as those in Table 1. Our observations reveal a significant discrepancy in the objective values between the training-free FreeDoM and the training-based PPAP [12], particularly at a lower number of steps (20 or 50 steps), which indicates slower convergence. However, when we incorporate the Polyak step size into FreeDoM, this convergence issue is substantially mitigated.

|  | RN-50 | RN-50+RA | Robust RN-50 | Real Data |
|---|---|---|---|---|
| RN-50 | 5.91 | 5.91 | 5.93 | 5.91 |
| Robust RN-50 | 6.17 | 5.98 | 5.93 | 5.93 |

Table 4: Quantitative experiments for the adversarial gradient. RN-50 stands for ResNet-50 and RA stands for random augmentation trick. Robust RN-50 is adversarial robust ResNet-50 from [35]. The columns represent different guidance networks.

|  | FreeDoM | FreeDoM + P | PPAP (Training-based) |
|---|---|---|---|
| DDIM-20 | 2439 | 2190 | 2032 |
| DDIM-50 | 1821 | 1635 | 1607 |
| DDIM-100 | 1657 | 1504 | 1509 |

Table 5: Quantitative experiments for the slower convergence. P stands for Polyak step size. The experimental setting follows the segmentation map guidance of Table 1.

### C.4  Efficiency of Random Augmentation

Given the multiple invocations of the guidance network necessitated by random augmentation (RA), concerns regarding the efficiency of this approach are understandable. However, it is important to note that, compared to the diffusion backbone, the guidance network exhibits a more lightweight architecture, thereby mitigating any significant increase in computational demand. To empirically illustrate this point, we present the computation times associated with varying degrees of augmentation in Table 6. In the conducted experiments, we set the cardinality of the set $\mathcal{T}$ to 10, thereby having little impact on the inference time. These experiments were conducted on a single NVIDIA A100 GPU.

| Setting | w/o RA | $|\mathcal{T}| = 1$ | $|\mathcal{T}| = 10$ | $|\mathcal{T}| = 20$ | $|\mathcal{T}| = 30$ |
|---|---|---|---|---|---|
| CLIP Score | 0.541 | 0.544 | 0.571 | 0.592 | 0.625 |

Table 6: Inference time (seconds per diffusion step) of different random augmentation configurations. The diffusion backbone is ImageNet diffusion and the guidance network is CLIP-B/16.

# D Proofs

## D.1 Definitions

This subsection introduces a few definitions that are useful in the following sections.

**Definition D.1.** (*L*-Lipschitz) A function $f : \mathbb{R}^n \to \mathbb{R}^m$ is said to be *L*-Lipschitz if there exists a constant $L \geq 0$ such that $\|f(\boldsymbol{x}_2) - f(\boldsymbol{x}_1)\| \leq L\|\boldsymbol{x}_2 - \boldsymbol{x}_1\|$ for all $\boldsymbol{x}_1, \boldsymbol{x}_2 \in \mathbb{R}^n$.

**Definition D.2.** (PL condition) A function $f : \mathbb{R}^n \to \mathbb{R}$ satisfies PL condition with parameter $\mu$ if $\|\nabla f(\boldsymbol{x})\|^2 \geq \mu f(\boldsymbol{x})$.

## D.2 Proof for Proposition 3.1

Denote $\hat{\boldsymbol{x}}_0 = \mathbb{E}_{p(\boldsymbol{x}_0|\boldsymbol{x}_t)}(\boldsymbol{x}_0)$, the gradient guidance term in (5) can be written as the following:

$$\nabla_{\boldsymbol{x}_t}\ell\left[f_\phi(\hat{\boldsymbol{x}}_0), \boldsymbol{y}\right] = \frac{\partial\ell}{\partial\hat{\boldsymbol{x}}_0}\nabla_{\boldsymbol{x}_t}\left(\frac{\boldsymbol{x}_t + \sigma_t^2\nabla_{\boldsymbol{x}_t}\log p_t(\boldsymbol{x}_t)}{\sqrt{\alpha_t}}\right) = \frac{\partial\ell}{\partial\hat{\boldsymbol{x}}_0}\frac{\text{Cov}[\boldsymbol{x}_0|\boldsymbol{x}_t]}{\sigma_t^2\sqrt{\alpha_t}}, \tag{9}$$

where the last equality follows the variance of Tweedie's formula $\text{Cov}[\boldsymbol{x}_0|\boldsymbol{x}_t] = \sigma_t^2(\boldsymbol{I} + \sigma_t^2\nabla^2\log p_t(\boldsymbol{x}_t))$ [10].

For the first condition, the Lipschitz constant satisfies

$$|\ell_t(\boldsymbol{x}_1) - \ell_t(\boldsymbol{x}_2)| \leq \frac{L_f}{\sqrt{\alpha_t}}\|\boldsymbol{x}_1 - \nabla_{\boldsymbol{x}_1}\log p_t(\boldsymbol{x}_1) - \boldsymbol{x}_2 + \nabla_{\boldsymbol{x}_t}\log p_t(\boldsymbol{x}_2)\|$$

$$\leq \frac{L_f(1 + L_p)}{\sqrt{\alpha_t}}\|\boldsymbol{x}_1 - \boldsymbol{x}_2\|,$$

and the PL constant satisfies

$$\ell_t(\boldsymbol{x}_t) \leq \frac{1}{\mu}\left\|\frac{\partial\ell}{\partial\hat{\boldsymbol{x}}_0}\right\|^2 = \frac{1}{\mu}\|\nabla_{\boldsymbol{x}_t}\ell\cdot\text{Cov}^{-1}(\boldsymbol{x}_0|\boldsymbol{x}_t)\sigma_t^2\sqrt{\alpha_t}\|^2 \leq \frac{\sigma_t^4\alpha_t}{\mu\lambda_{min}^2}\|\nabla_{\boldsymbol{x}_t}\ell\|^2.$$

The second and third conditions directly follow Lemma D.3.

**Lemma D.3.** *(Linear Convergence Under PL condition; [22]) Denote $\boldsymbol{x}^0$ as the initial point and $\boldsymbol{x}^t$ as the point after $t$ gradient steps. If the function is L-Lipschitz and $\mu$-PL, gradient descent with a step size $\eta = \frac{1}{L}$ converges to a global solution with $\ell(\boldsymbol{x}^t) \leq (1 - \mu/L)^t\ell(\boldsymbol{x}^0)$.*

## D.3 Proof for Proposition 3.2

*Proof.* The proof of this theorem is based on the proof of Lemma 1 in [34]. By the definition of expectation, we have

$$\hat{f}(\boldsymbol{x}) = \mathbb{E}_{\boldsymbol{\epsilon}\sim\mathcal{N}(0,\boldsymbol{I})}[f(\boldsymbol{x} + \sigma_t\boldsymbol{\epsilon})] = (f \circledast \mathcal{N}(0, \sigma_t^2\boldsymbol{I}))(\boldsymbol{x})$$

$$= \frac{1}{(2\pi\sigma_t^2)^{n/2}}\int_{\mathbb{R}^n} f(\boldsymbol{z})\exp\left(-\frac{1}{2\sigma_t^2}\|\boldsymbol{x} - \boldsymbol{z}\|^2\right)\mathrm{d}\boldsymbol{z}.$$

We then show that for any unit direction $\boldsymbol{u}$, $\boldsymbol{u}^T\nabla\hat{f}(\boldsymbol{x}) \leq \sqrt{\frac{2}{\pi\sigma_t^2}}$. The derivative of $\hat{f}$ is given by

$$\nabla\hat{f}(\boldsymbol{x}) = \frac{1}{(2\pi\sigma_t^2)^{n/2}}\int_{\mathbb{R}^n} f(\boldsymbol{z})\nabla\exp\left(-\frac{1}{2\sigma_t^2}\|\boldsymbol{x} - \boldsymbol{z}\|^2\right)\mathrm{d}\boldsymbol{z}$$

$$= \frac{1}{(2\pi\sigma_t^2)^{n/2}\sigma_t^2}\int_{\mathbb{R}^n} f(\boldsymbol{z})(\boldsymbol{x} - \boldsymbol{z})\exp\left(-\frac{1}{2\sigma_t^2}\|\boldsymbol{x} - \boldsymbol{z}\|^2\right)\mathrm{d}\boldsymbol{z}.$$

Thus, the Lipschitz constant is computed as

$$\boldsymbol{u}^T\nabla\hat{f}(\boldsymbol{x}) \leq \frac{C}{(2\pi\sigma_t^2)^{n/2}}\int_{\mathbb{R}^n}|\boldsymbol{u}^T(\boldsymbol{x} - \boldsymbol{z})/\sigma_t^2|\exp\left(-\frac{1}{2\sigma_t^2}\|\boldsymbol{x} - \boldsymbol{z}\|^2\right)\mathrm{d}\boldsymbol{z}$$

$$= \frac{C}{(2\pi\sigma_t^2)^{1/2}}\int_{-\infty}^{\infty}|s|\exp\left(-\frac{1}{2}s^2\right)\mathrm{d}s = \sqrt{\frac{2}{\pi\sigma_t^2}}.$$

Similarly, for the Lipschitz constant of the gradient, we have

$$\|\nabla^2 \hat{f}(\boldsymbol{x})\|_{\mathrm{op}} \leq \|\nabla^2 \hat{f}(\boldsymbol{x})\|_2$$
$$\leq \frac{C}{\sqrt{2\pi}\sigma_t \cdot \sigma_t^4} \left( \int_{-\infty}^{\infty} s^2 \exp(-\frac{1}{2}s^2/\sigma_t^2)\mathrm{d}s + \int_{-\infty}^{\infty} \sigma_t^2 \exp(-\frac{1}{2}s^2/\sigma_t^2)\mathrm{d}s \right) = \frac{2C}{\sigma_t}.$$

$\square$

### D.4 Proof for Proposition 3.3

*Proof.* We first analyze the discretization error of a single step from time $s$ to $t$. We denote the update variable for DDIM is $\boldsymbol{x}_t^*$ and the optimal solution of the diffusion ODE at time $t$ as $\boldsymbol{x}_t^*$. Let $\sigma_t = \sqrt{1 - \alpha_t}$, $\lambda_t = \frac{1}{2}\log(\frac{\alpha_t}{1-\alpha_t})$ and $h = \lambda_t - \lambda_s$. According to (B.4) of [26], the update of DDIM solver is given by

$$\boldsymbol{x}_t = \sqrt{\frac{\alpha_t}{\alpha_s}}\boldsymbol{x}_s - \sigma_t(e^h - 1)u(\boldsymbol{x}_s, s). \tag{10}$$

A similar relationship can be obtained for the optimal solution

$$\boldsymbol{x}_t^* = \sqrt{\frac{\alpha_t}{\alpha_s}}\boldsymbol{x}_s^* - \sigma_t(e^h - 1)u(\boldsymbol{x}_s^*, s) + O(h^2). \tag{11}$$

We then bound the error between $\boldsymbol{x}_t$ and $\boldsymbol{x}_t^*$. We have

$$\|\boldsymbol{x}_t^* - \boldsymbol{x}_t\| \leq \sqrt{\frac{\alpha_t}{\alpha_s}}\|\boldsymbol{x}_s^* - \boldsymbol{x}_s\| + \sigma_t(e^h - 1)\|u(\boldsymbol{x}_s^*, s) - u(\boldsymbol{x}_s, s)\| + O(h^2)$$

$$\leq \sqrt{\frac{\alpha_t}{\alpha_s}}\|\boldsymbol{x}_s^* - \boldsymbol{x}_s\| + \sigma_t(e^h - 1)L\|\boldsymbol{x}_s^* - \boldsymbol{x}_s\| + O(h^2)$$

$$\leq O((1 + L)\|\boldsymbol{x}_s^* - \boldsymbol{x}_s\|) + O(h^2)$$

If we run DDIM for $M$ steps, $h_{\max} = O(1/M)$, and we achieve the discretization error bound for DDIM algorithm:

$$\|\boldsymbol{x}_0 - \boldsymbol{x}_0^*\| \leq O(M(1 + L)^M h_{\max}^2) = \boldsymbol{x}_0^* + O((1 + L^M)/M).$$

$\square$

### D.5 Proof for Proposition 4.1

*Proof.* By the definition of expectation, we have

$$\hat{f}(\boldsymbol{x}) = \mathbb{E}_{\boldsymbol{\epsilon} \sim p(\boldsymbol{\epsilon})}[f(\boldsymbol{x} + \sigma_t\boldsymbol{\epsilon})] = (f \circledast p)(\boldsymbol{x}) = \int_{\mathbb{R}^n} f(\boldsymbol{z})p(\boldsymbol{x} - \boldsymbol{z})\mathrm{d}\boldsymbol{z}.$$

Then we compute the Lipschitz constant

$$\boldsymbol{u}^T\nabla\hat{f}(\boldsymbol{x}) \leq C \int_{\mathbb{R}^n} \|\nabla p(\boldsymbol{x} - \boldsymbol{z})\|_2 \mathrm{d}\boldsymbol{z} = C \int_{\mathbb{R}^n} \|\nabla p(\boldsymbol{z})\|_2 \mathrm{d}\boldsymbol{z}.$$

As for the gradient Lipschitz constant, we have

$$L = \|\nabla^2 \hat{f}(\boldsymbol{x})\|_{\mathrm{op}} \leq C \left\| \int_{\mathbb{R}^n} \nabla^2 p(\boldsymbol{x} - \boldsymbol{z})\mathrm{d}\boldsymbol{z} \right\|_{\mathrm{op}} \leq C \int_{\mathbb{R}^n} \|\nabla^2 p(\boldsymbol{z})\|_{\mathrm{op}}\mathrm{d}\boldsymbol{z},$$

where $\| \cdot \|_{\mathrm{op}}$ is the operator norm of a matrix. $\square$

## E More Qualitative Results

### E.1 CelebA-HQ

### E.2 ImageNet

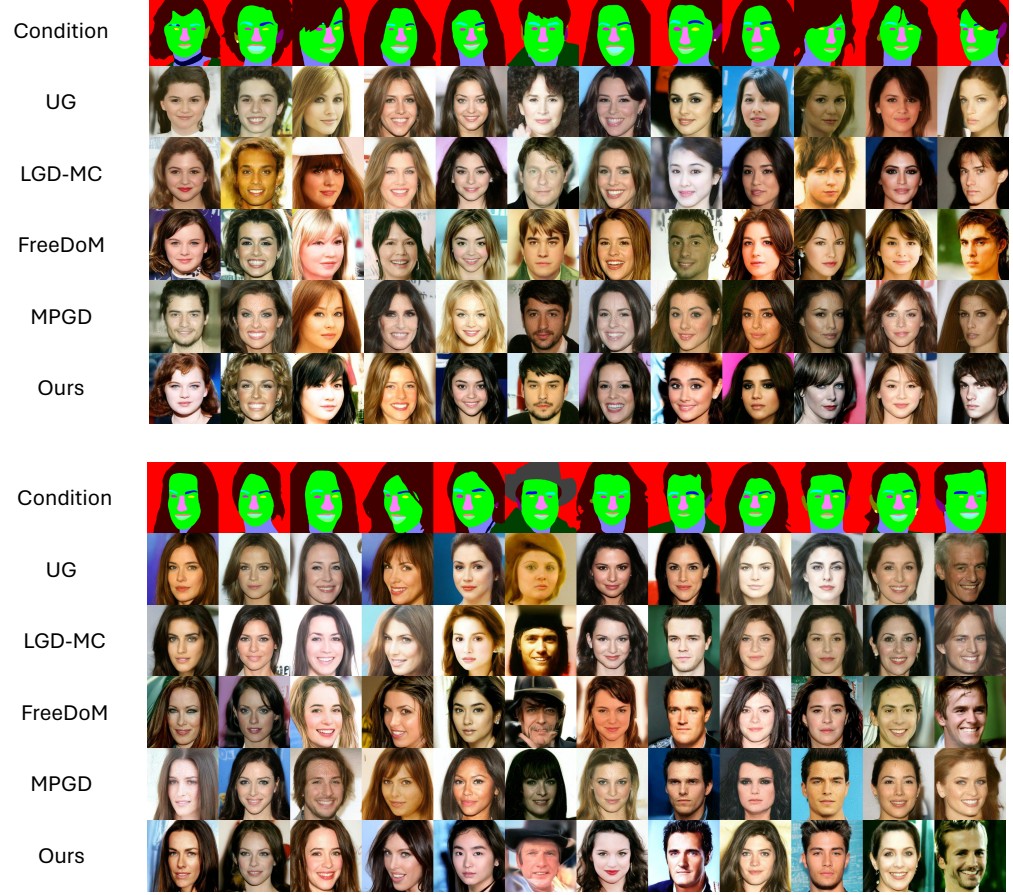

Figure 8: More qualitative results of CelebA-HQ with zero-shot segmentation guidance. The images are randomly selected.

## E.3 Human Motion

See "TeX Source/videos.pptx".

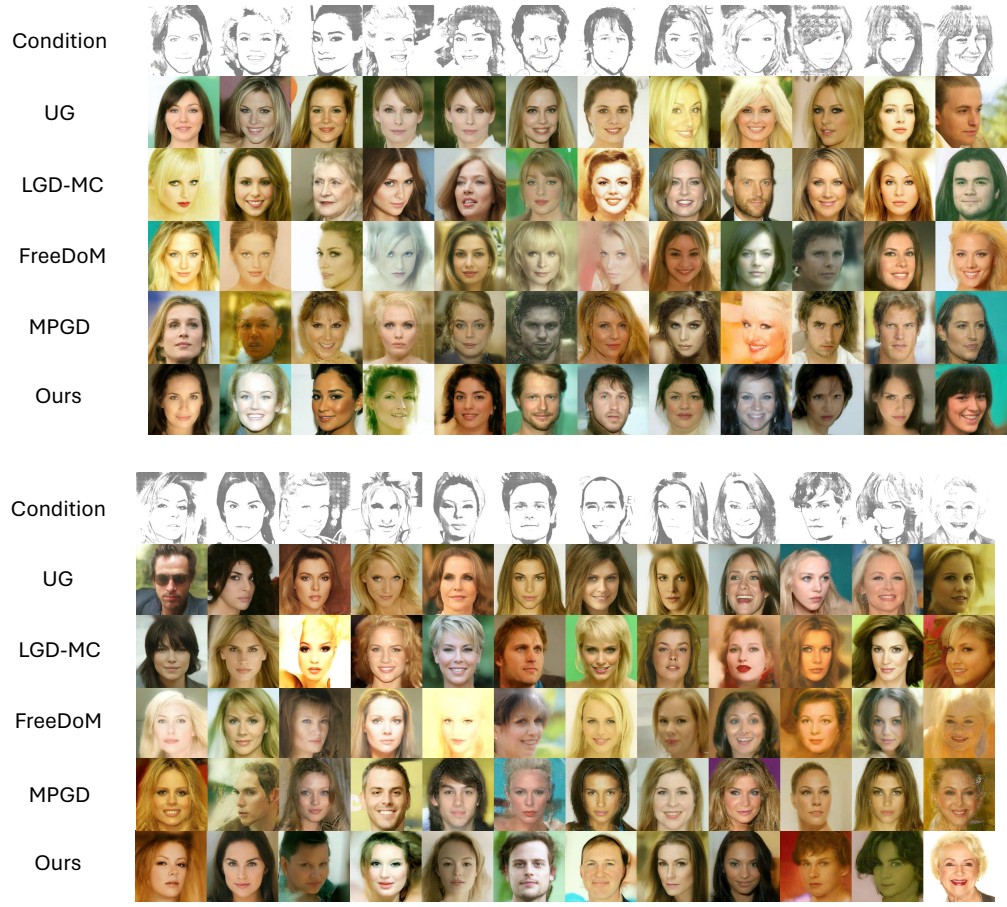

Figure 9: More qualitative results of CelebA-HQ with zero-shot sketch guidance. The images are randomly selected.

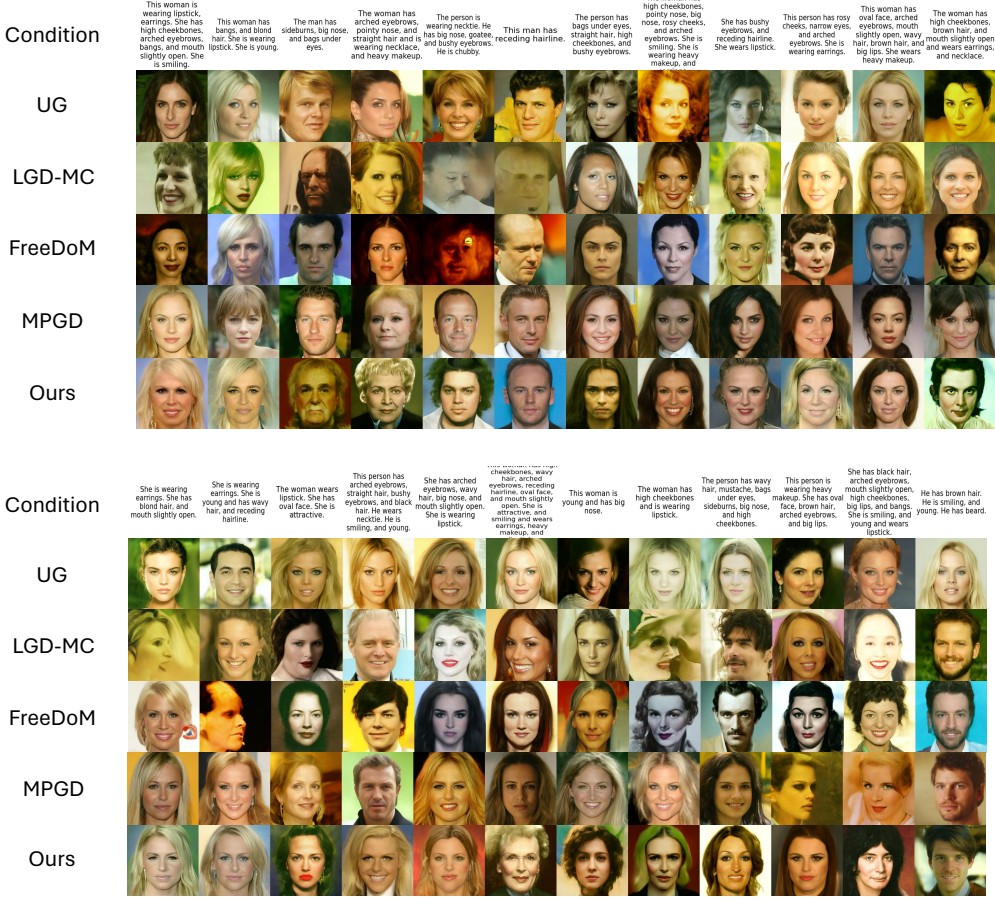

Figure 10: More qualitative results of CelebA-HQ with zero-shot text guidance. The images are randomly selected.

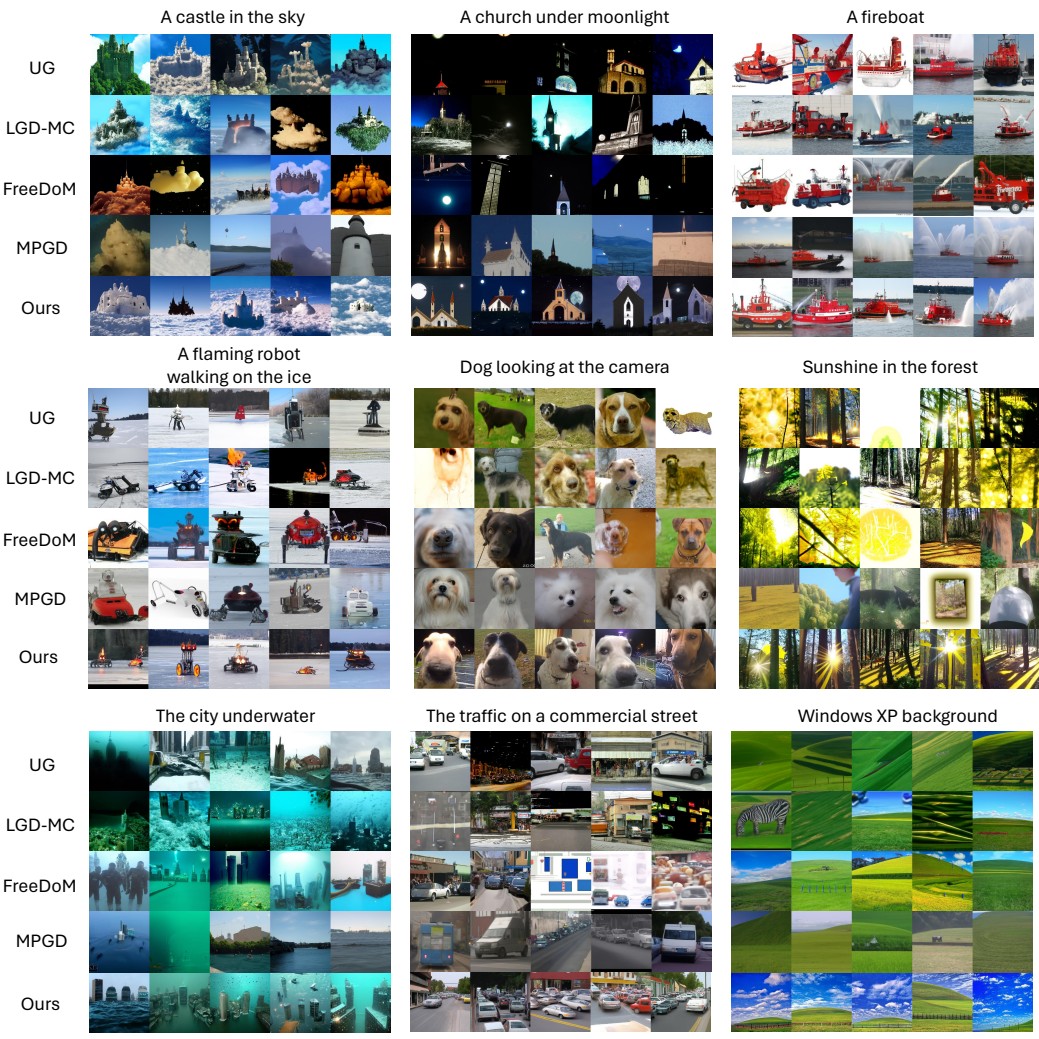

Figure 11: More qualitative results of ImageNet with zero-shot text guidance. The images are randomly selected.

