# OpenReview forum: "Understanding and Improving Training-free Loss-based Diffusion Guidance"
_NeurIPS.cc/2024/Conference — NeurIPS 2024 poster_

### Official Review · Reviewer_uPWj · 2024-06-30

**Soundness:** 2
**Presentation:** 3
**Contribution:** 2
**Rating:** 5
**Confidence:** 5

**Summary:**

This paper explores training-free loss-based diffusion guidance, providing an overview of how training-free guidance functions from an optimization perspective. It analyzes the challenges of training-free guidance, particularly regarding adversarial gradient issues and slower convergence rates. Additionally, the paper proposes two improvements: random augmentation and Polyak step size. These enhancements are validated through experiments across various applications.

**Strengths:**

- The theoretical analysis offers valuable insights into understanding training-free guidance.

- The proposed tricks, random augmentation and Polyak step size, demonstrate effectiveness.

**Weaknesses:**

Although the paper provides a thorough theoretical analysis of why training-free guidance works and doesn’t work, the empirical evidence is insufficient to fully support the claims. The propositions are based on the assumption of Lipschitz continuity, which may not hold for complex deep neural networks, therefore, a more comprehensive quantitative analysis is necessary. Here are my detailed comments:

- The experiment presented in Figure 1 is somewhat confusing. The authors aim to convey three points: (1) the loss curve will oscillate initially and decrease eventually; (2) the guidance weight needs careful selection; (3) adversarial gradients may cause incorrect generated images despite low loss. The first point requires more extensive experiments rather than just two cases. The second point is not clearly inferred from the figure.
- The two limitations of training-free guidance, adversarial gradient, and slower convergence rates, lack quantitative experimental evidence.

Regarding the proposed method, while benchmarking many baselines on various tasks is appreciated, several issues remain unaddressed:

- The difference between UG and FreeDoM in this paper appears to be the selection of diffusion steps for the time-travel trick. However, UG also uses a “backward universal guidance” technique as mentioned in section 3.2 of the original paper [1].
- The authors did not choose classifier-based tasks, which is curious given that the paper analyzes the inferiority of training-free guidance compared to training-based guidance through this kind of task in Sec 3. Comparative experiments showing the gap between these approaches are essential. Including classifier-guided generation tasks would be beneficial to assess the significance of the two tricks’ improvements, and training-based baselines can be included in this task.
- The selection of hyper-parameters is concerning. With numerous hyper-parameters involved, how are the optimal values (e.g., step size, LGD-MC Monte Carlo samples, diffusion step range for time-travel) chosen? How is it ensured that baselines are implemented optimally?
- The random augmentation trick may have limitations:
  - While image augmentation techniques are mature, effective augmentation tricks for other domains (e.g., text, AI4Science, offline RL) may not be available, limiting the trick’s application.
  - Despite the claim that random augmentation does not introduce significant computational overhead, this may not hold true when using a large guidance network, as would be the case with a large, generalist foundation model for scoring samples.



[1] UNIVERSAL GUIDANCE FOR DIFFUSION MODELS. https://openreview.net/pdf?id=pzpWBbnwiJ

**Questions:**

- What is the rationale behind choosing the Polyak step size? It appears to be a post-hoc decision without strong intuition.
- Since random augmentation is omitted in MDM, is the only difference between “Ours” and “FreeDoM” the Polyak step size? The loss values in Table 3 show a significant difference between FreeDoM and Ours, and clarification on this point is needed.
- What's the value of $\eta$ set for DDIM?

**Limitations:**

Yes.

---

> ### Author Rebuttal · Authors · 2024-08-07
>
> We sincerely thank the reviewer for your insightful and constructive feedbacks. We have added more experiments, baselines, and ablation studies.
>
> > Weakness 1: Assumption of Lipschitz continuity
>
> The assumption of Lipschitz continuity is a standard prerequisite for analyzing diffusion models (e.g., [A]). Without this assumption, constructing a similar argument becomes significantly more challenging (comparing [A] to [B]). Additionally, this assumption is frequently utilized in the analysis of neural networks (e.g., [C]). Therefore, we have incorporated this assumption to facilitate deeper insights while maintaining a streamlined proof structure.
>
> > Weakness 2: The experiment presented in Figure 1 is somewhat confusing.
>
> Thank you for your assistance in clarifying Proposition 3.1 and Figure 1. The outcomes (1) and (2) are direct consequences of Proposition 3.1, and Figure 1 serves merely as an illustration of the first claim. We acknowledge the reviewer's point that verifying the first claim necessitates more experiments, and we have included **additional experiments in Figure 1 within the PDF of the global rebuttal**. Claim (3) is discussed in Section 3.2, and we will ensure to revise this paragraph in the updated manuscript accordingly.
>
> > Weakness 3: Quantitative experimental evidence of two drawbacks.
>
> The qualitative evidence supporting the presence of adversarial gradients and slower convergence rates is presented in Figures 2 and 3, respectively. We concur with the reviewer on the importance of quantitative evidence and **we include quantitative experiments in Table 3, 4 in PDF of global rebuttal**. To evaluate adversarial gradients, we employ a classifier setting with pre-trained adversarially robust networks and test on 1,000 generated images. When utilizing ResNet-50 as the guidance network, it minimizes its own loss, yet the loss on a Robust ResNet-50 remains high, indicating the existence of adversarial gradients. Our proposed random augmentation method is effective in mitigating these adversarial gradients.
>
> Regarding the slower convergence, the simplicity of the classifier setting may obscure this observation. Therefore, we shifted our focus to the more challenging segmentation map guidance. Here, we observed a noticeable gap in the objective values between the training-free FreeDoM and the training-based PPAP [D] when the number of steps is small, demonstrating the slower convergence. The implementation of the Polyak step size helps to narrow this gap.
>
>
> > Weakness 4: Description of UG
>
> We thank the reviewer for correcting this point and we will modify the description of UG in Appendix.
>
> > Weakness 5: Classifier-based tasks and ablation studies
>
> Thank you for your valuable suggestion. Similar to FreeDoM and LGD-MC, our work is concentrated on tasks where training-based methods have yet to be fully developed. Given that both FreeDoM and LGD-MC have demonstrated commendable performance on these benchmarks, we believe that surpassing these established baselines is indicative of our method's strong performance.
>
> We concur with the reviewer on the importance of incorporating training-based methods and conducting ablation studies. To address this, we have included training-based methods and an ablation study of our approach in **Table 2 of the PDF in the global rebuttal**. Polyak step size expedites convergence and significantly enhances the objective value. While random augmentation does improve the Fidelity (FID), it can adversely affect the objective value due to the introduction of randomness. When these two elements are combined, they yield the best performance.
>
> > Weakness 6: Hyperparameter choice
>
> Regarding the step size selection, we initially explored a range from $10^{-5}$ to $10^3$. If a particular step size $\lambda$ yields valid images, we then refine our search within the interval [$\lambda$, $10\lambda$] using 20 evenly spaced values. In the case of Monte Carlo samples, the LGD-MC study noted negligible differences between using $n=10$ and $n=100$, and thus we adopt $n=10$. We sourced the implementations of UG, FreeDoM, and MPGD directly from their respective GitHub repositories.
>
> > Weakness 7: Limitations of random augmentation
>
> Regarding the first limitation, there exist data augmentation methods within the domains of molecular [A] and offline reinforcement learning [B]. For the second limitation, we concur with the reviewer that the adoption of a foundation model may result in increased computation time. However, it is important to note that random augmentation processes are highly parallelizable, and the guidance models utilized for current training-free guidance are not so large. We have provided detailed timings for the CLIP model in Table 4 of the original paper.
>
> > Question 1: Rationale behind Polyak
>
> Training-free guidance is indeed challenged by a slow convergence issue, as outlined in Proposition 3.3. The simplest remedy for this is to adjust the step size. The Polyak step size has been demonstrated to be optimal under a variety of conditions [13], which is why we have chosen to implement it in our approach.
>
> > Question 2: Clarification on MDM
>
> Unlike image generation, motion diffusion involves creating $192$ frames of motion where consistency between neighboring frames is crucial to produce a viable trajectory. This requirement significantly complicates the optimization challenge, making the selection of an appropriate step size even more critical. Due to this complexity, MPGD struggles to generate a feasible motion, as detailed in Appendix B.2.
>
> > Question 3: the value of $\eta$
>
> The value of $\eta$ is set the same as the released code of these baselines.
>
>
> [A] AugLiChem: data augmentation library of chemical structures for machine learning. MLST 2022.
>
> [B] Reinforcement learning with augmented data. NeurIPS 2020.
>
> [C] On the spectral bias of neural networks. ICML 2019.
>
> [D] Towards Practical Plug-and-Play Diffusion Models, CVPR 2023.

---

> > ### Author Response · Authors · 2024-08-09
> >
> > We apologize for the oversight in not including references for Weakness 1 in our previous submission. The references for the rebuttal of Weakness 1 are as follows:
> >
> > [A] DPM-Solver: A Fast ODE Solver for Diffusion Probabilistic Model Sampling in Around 10 Steps
> >
> > [B] Accelerating convergence of score-based diffusion models, provably

---

> > > ### Comment · Reviewer_uPWj · 2024-08-13
> > >
> > > Thank you to the authors for the detailed responses. I have some follow-up comments:
> > >
> > > 1. I apologize for not fully understanding the meaning of Table 3 and Table 4 in the uploaded PDF. Could you please clarify what the values in these tables represent? I would appreciate a more detailed explanation of the settings for each table.
> > >
> > > 2. Regarding the classifier-based tasks, I appreciate that the authors chose to evaluate their method on tasks where FreeDoM and LGD-MC have shown strong performance. However, I still believe that using tasks involving classifier guidance would be more appropriate. This would allow for more direct comparisons with classifier-based methods (or even classifier-free guidance), making both quantitative comparisons and qualitative analyses easier.
> > >
> > > 3. Thank you for highlighting the presence of augmentation methods in molecule and RL domains. My concern is that it's difficult to justify the effectiveness of all augmentation methods in training-free, loss-based diffusion guidance, as the efficacy of augmentation can vary significantly across different domains.

---

> ### Author Response · Authors · 2024-08-13
>
> We greatly appreciate your insightful suggestions and comments, which have significantly contributed to the refinement of our paper. We are also thankful for your acknowledgment of our rebuttal efforts.
>
> **Comment 1:** Regarding Table 3, it is designed to evaluate the adversarial gradient issue in training-free guidance. To assess the presence of adversarial gradients, we utilized an adversarially robust ResNet-50 [33], which is trained to resist adversarial gradients. Since ResNet-50 serves as our guidance network, a large loss gap between the standard ResNet-50 and the robust variant indicates a severe adversarial gradient issue. In Table 3, the values represent the loss value. The columns denote the guidance network type, while the rows indicate the loss tested on either ResNet-50 or Robust ResNet-50. The values reported are the average over 1000 images, providing a quantitative evaluation. When using ResNet-50 for guidance, its loss on ResNet-50 is as low as that of real images from the same class (value 5.91). However, the loss for Robust ResNet-50 is significantly higher (value 6.17), suggesting susceptibility to adversarial gradients. By employing ResNet-50 with random augmentation for guidance, we observe a marked reduction in this gap (from 5.91 to 5.98), underscoring the effectiveness of our proposed method.
>
> Table 4 aims to compare the convergence of training-free guidance with training-based guidance, specifically PPAP [D]. Given that classifier guidance presents a simple scenario and all methods converge relatively fast, it tends to obscure the observations. Consequently, we have directed our attention towards the more complex task of segmentation map guidance in Table 1. The columns in Table 4 list different methods, while the rows indicate the sampling steps. The values represent the "distance" mentioned in Table 1 and are averaged over 1000 images under the same conditions as those in Table 1. Our observations reveal a significant discrepancy in the objective values between the training-free FreeDoM and the training-based PPAP [D], particularly at a lower number of steps (20 or 50 steps), which indicates slower convergence. However, when we incorporate the Polyak step size into FreeDoM, this convergence issue is substantially mitigated.
>
> **Comment 2:** As for the second concern, we are thankful for your acknowledgment of our efforts in the rebuttal. We have also conducted comparisons with training-based methods, as illustrated by PPAP in Table 1 of the PDF provided in the global rebuttal. PPAP extends classifier guidance to accommodate a broader range of conditions by fine-tuning the pretrained model on clean images to create different experts, each responsible for a specific range of noise level. The gradients from the corresponding experts are then used for guidance at corresponding noise levels. To optimize performance of PPAP, we have set the number of experts to $10$, which is the maximum reported in [D]. As indicated in Table 1, our proposed method shows performance on par with the training-based PPAP. With respect to the reviewer's request for additional experiments under class conditions, we are currently conducting these experiments. However, due to the high number of required samples and the author-reviewer discussion coming to an end, it is possible that they cannot be completed before the reviewer-author discussion period closes. We assure you that the results of these experiments will be included in the revised manuscript.
>
> **Comment 3:** Regarding the third question, the effectiveness of random augmentation is supported by Proposition 4.1. This proposition does not rely on the assumption of Lipschitz continuity or any specific distribution for the augmentation, suggesting its broad applicability across various modalities. Consequently, the theoretical foundation guarantees the effectiveness of random augmentation in AI4Science or RL without imposing any strong assumptions. As we approach the end of the author-reviewer discussion and considering AI4Science or RL represents a novel application setting for our paper, it is regrettably unlikely that additional experiments can be completed prior to the submission deadline. Nonetheless, we are committed to including these experiments in the revised manuscript and we appreciate the reviewer's understanding in this matter.

---

> > ### Comment · Reviewer_uPWj · 2024-08-13
> >
> > Thanks for the timely response. Considering of the theoretical contribution of this paper which helps to understand training-free loss-based diffusion guidance better, I raise the score to a 5.

---

### Official Review · Reviewer_DfoY · 2024-07-12

**Soundness:** 3
**Presentation:** 2
**Contribution:** 3
**Rating:** 5
**Confidence:** 4

**Summary:**

This paper explores the theoretical aspects of training-free loss-based guidance mechanisms in diffusion models and improves them from theoretical findings. The first results are in explaining why the guidance strength depended on $\sqrt{\alpha_t}$ worked well in FREEDOM. Then, the authors explore the adversarial gradients, which result in a lower loss but produce generations with the incorrect condition. Specifically, they demonstrated that a time-dependently trained network with Gaussian noises is smoother compared to a general network. This smoothness can cause a slowdown of guidance. From these theoretical results, they propose to utilize random augmentation to increase the smoothness of the off-the-shelf network and improve the convergence speed with Polyak step size, which accelerates convergence by adaptive gradient step size. Experimental results show that their method can outperform loss-based guidance methods including Universal Guidance, Loss-Guided Diffusion, FreeDom, and MPGD-Z.

**Strengths:**

- This paper explores the theoretical aspects of training-free loss-based guidance. The authors explain well-known practices in previous works with theoretical aspects, such as FreeDom.
- Various benchmark datasets are utilized for evaluating their method.

**Weaknesses:**

- My first major concern is missed baselines and related works. This work aims to improve off-the-shelf models’ guidance with the training-free based method, and [A] deals with the same configuration. [A] elucidates the design space of off-the-shelf guidance, and smoothed classifier, joint classifier guidance, and guidance schedules are explored in their work. I think that the smoothed classifier concept in [A] is very similar to random augmentation for the smoother network in this work, and similar guidance schedule schemes are explored in [A] and this work. Therefore, [A] should be discussed in this work and be compared in the experimental section. Also, I found [C] and [D].
- In the other line of work, [B] proposes the utilization of off-the-shelf models by finetuning that model to be the time-dependent network. Although [B] can be categorized into the training-required method, it would be better to discuss this kind of training-required method and compare it in the experimental section. Showing the performance gaps between training-free and training-needed methods is needed to validate the effectiveness of the proposed method.
- [A] and [B] choose the imagenet classifier guidance with imagenet trained diffusion models and various diffusion models use imagenet class conditional generation as the core benchmark. However, this work does not incorporate this major benchmark in their benchmark, so it can make reader hard to recognize their absolute performance.
- There are no details about the Polyak Step size, and if Algorithm 2 is all of the details of the Polyak Step Size, there have been many works to utilize normalized gradients and I felt that this is not novel.

**Questions:**

## Typo
- Line 875 in Appendix: $||f(x_2)-f(x_2)||$ ->  $||f(x_2)-f(x_1)||$
- Equations are referred to as (number). I am very confused to understand whether (number) is an equation or a bullet point. How about using Eq. (number)?


## References

- [A] ELUCIDATING THE DESIGN SPACE OF CLASSIFIER GUIDED DIFFUSION GENERATION, ICLR 2024

- [B] Towards Practical Plug-and-Play Diffusion Models, CVPR 2023.

- [C] ADJOINTDPM: ADJOINT SENSITIVITY METHOD FOR GRADIENT BACKPROPAGATION OF DIFFUSION PROBABILISTIC MODELS, ICLR 2024.

- [D] Towards Accurate Guided Diffusion Sampling through Symplectic Adjoint Method, Arxiv 2023.

**Limitations:**

The higher number of NFEs required in training-free guidance and adversarial guidance are mentioned in the limitation section.

---

> ### Author Rebuttal · Authors · 2024-08-07
>
> We sincerely thank the reviewer for your insightful and constructive feedbacks. Based on the comments, we have added more baselines.
>
> > Weakness 1 and 2: Comparison with [A,B,C,D]
>
> We appreciate the reviewers for highlighting these pertinent references. We have now included [B] (training-based PPAP) and [D] (SAG) as additional baselines, with the results presented in Table 1 of the PDF for global rebuttal. Regarding hyperparameters, we have opted for a solve step of $n=4$ and selected the number of experts in PPAP to be $10$, consistent with the original papers.
>
> [A] primarily focuses on employing a classifier as the guidance network. For instance, the adoption of a joint direction, a softplus classifier, and the adjustment of classifiers' temperature are approaches that are difficult to be applied to general loss functions. Since [D] builds upon the general methodology outlined in [C], we have chosen to include only [D] in our analysis. Our findings indicate that SAG outperforms FreeDoM, particularly in terms of the objective value, aligning with the results reported in the SAG paper. However, it is surpassed by our proposed method. Moreover, the performance of our method is comparable to that of training-based approaches.
>
> We would also like to clarify the differences between our work and [A]. Firstly, regarding the smooth classifier, the methodology employed in [A] involves using a softplus classifier and adjusting the classifier's temperature, which may not be readily adaptable for general conditions. In contrast, the random augmentation method we propose is independent of the loss function and guidance network. From a motivational standpoint, our paper introduces an additional significant motivation: to eliminate the adversarial gradient, an aspect that has been overlooked by existing studies. Secondly, concerning the scheduler, our paper proposes a step size choice that is orthogonal to the scheduler design.
>
> > Weakness 3: Class condition as benchmark
>
> Similar to FreeDoM and LGD-MC, our work concentrates on tasks for which training-based methods are not yet fully developed. We have utilized the same benchmarks as those employed in FreeDoM and LGD-MC. Given that both FreeDoM and LGD-MC have demonstrated commendable performance on these benchmarks, we believe that surpassing these established baselines is indicative of our method's robust performance. We are open to different viewpoints and welcome the opportunity for further discussion.
>
> > Weakness 4: Polyak Step size
>
> We are grateful for the opportunity to clarify the novelty of our approach. Due to space constraints in the paper, we were unable to delve into the specifics of the Polyak step size, which may have led to some misunderstanding. The Polyak step size is a method of choosing step sizes, distinct from normalization, and it unifies various step size choices with a proven optimal convergence rate under a range of conditions [13]. In contrast, normalization methods like those used in other works (e.g., DSG) typically focus on projection to the manifold. It's important to note that the Polyak step size and manifold projections are orthogonal concepts [E], and their combination in our method is more than a mere amalgamation of the two methods' codes. Furthermore, we have compared our method against the state-of-the-art DSG [F], as shown in the global response PDF, and our proposed method demonstrates superior performance.
>
> > Typo
>
> Thank you for your careful reading. We will carefully correct the typos in the revised manuscripts.
>
> [E] Iteration-complexity of the subgradient method on Riemannian manifolds with lower bounded curvature. Optimization 2019.
>
> [F] Guidance with spherical gaussian constraint for conditional diffusion. arXiv:2402.03201.

---

> > ### Comment · Reviewer_DfoY · 2024-08-09
> >
> > My concerns are fully addressed. Accordingly, I raised my rating as 5

---

> > > ### Author Response · Authors · 2024-08-09
> > >
> > > Thank you for your thorough review and insightful suggestions. Your comments have been invaluable in refining our paper. We appreciate your positive reception of our rebuttal and are glad that our response has addressed your concerns. We are committed to carefully revising the manuscript according to your feedback.

---

### Official Review · Reviewer_P7k4 · 2024-07-12

**Soundness:** 3
**Presentation:** 3
**Contribution:** 2
**Rating:** 5
**Confidence:** 4

**Summary:**

This paper examines the mechanisms and limitations of training-free guidance for diffusion models and develops a collection of techniques to overcome the limitations accompanied by both theoretical and empirical results.

**Strengths:**

1.	This paper performs the theoretical analysis on the training-free diffusion model from the optimization perspective, which presents the problems and drawbacks in training-free guidance.
2.	This paper also proposes corresponding solutions for the two issues in training-free guidance: random augmentation for the adversarial gradient, Polyak step size for improving the convergence.

**Weaknesses:**

1.	The paper misses the recent literature in training-free diffusion model, such as RED-diff [1] and DSG [2].
2.	Although the theoretical analysis in this paper illustrates the problems in training-free guidance, the proposed solutions are not particularly novel. Similar ideas have been applied in previous works such as LGD-MC and DSG.

[1] Mardani M, Song J, Kautz J, et al. A variational perspective on solving inverse problems with diffusion models[J]. arXiv preprint arXiv:2305.04391, 2023.

[2] Yang L, Ding S, Cai Y, et al. Guidance with spherical gaussian constraint for conditional diffusion[J]. arXiv preprint arXiv:2402.03201, 2024.

**Questions:**

1.	How is the performance of MPGD and LGD-MC modified with the random augmentation and Polyak step size?
2.	How is the proposed method compared with DPS, DSG and RED-diff?

**Limitations:**

The authors have clearly presented the limitations in the paper.

---

> ### Author Rebuttal · Authors · 2024-08-07
>
> We sincerely thank the reviewer for your insightful and constructive feedbacks. We have added the baselines and integrated the proposed methods into MPGD and LGD-MC.
>
> > Question 1: MPGD and LGD-MC with random augmentation and Polyak step size
>
> We are grateful for your suggestions to enhance our experimental section. As requested, **we have included additional experiments in Table 1 in PDF file of the global rebuttal**. In these experiments, when integrating LGD-MC, we set the number of random augmentations to 5 and the Monte Carlo iterations to 2. The results show that both LGD-MC and MPGD significantly outperform their original counterparts when combined with random augmentation and Polyak step size adjustments. This indicates that our proposed methods can complement existing techniques. Through further analysis, we determined that the Polyak step size is the primary factor contributing to these improvements. Notably, MPGD-Z, when used in conjunction with our approach, delivers exceptional performance, even surpassing the training-based PPAP [B].
>
> > Weakness 1 and Question 2: Comparison with DPS, DSG, and RED-diff
>
> Thank you for highlighting the necessity of including additional baselines. We have conducted further experiments with the DSG algorithm and presented the results in Table 1. It is important to clarify that the DPS algorithm is essentially the FreeDoM method, minus the time-travel techniques. Since DPS is analogous to FreeDoM without the time-travel aspect, we believe that the performance of FreeDoM can be indicative of DPS's performance. As for RED-diff, it employs diffusion as a form of regularization for the least squares problem. It is more suitable for linear inverse problems rather than conditional generation tasks. In our testing, RED-diff failed to generate valid images under conditions such as segmentation, sketch, or CLIP guidance. Consequently, we have decided not to include RED-diff in our baseline comparisons.  **We have incorporated the DSG results into Table 1 in the PDF file of the global rebuttal**. While DSG shows an improvement over FreeDoM, it still falls short when compared to our proposed methods.
>
> > Weakness 2: Proposed solution is not novel compared with LGD-MC and DSG.
>
> We appreciate your assistance in elucidating the distinct contributions of our work. The approaches of LGD-MC and DSG differ in their motivations and are orthogonal in their methodologies. In LGD-MC, the MC method is utilized to approximate the posterior distribution, which necessitates that the variance of the Gaussian noise be proportional to $\sigma_t/\sqrt{1 + \sigma_t^2}$. On the other hand, the rationale behind employing random augmentation is to mitigate adversarial gradients, allowing for the consistent application of the same augmentation across different steps. It is worth noting that random augmentation and MC can be employed concurrently, although this may result in an increased sampling time. The essence of DSG lies in the projection onto the manifold, a process that is orthogonal to the choice of step size. Moreover, various manifold projections can be seamlessly integrated with the Polyak step size, as detailed in [A].
>
> [A] Iteration-complexity of the subgradient method on Riemannian manifolds with lower bounded curvature. Optimization 2019.
>
> [B] Towards Practical Plug-and-Play Diffusion Models. CVPR 2023.

---

> > ### Comment · Reviewer_P7k4 · 2024-08-09
> >
> > Thanks for your responses and the efforts on the additioanl experiments. Although the theoretical contribution of this work should be acknowledged, the novelty of this paper is still limited in the reviewer's opinion. In that case, I am sticking to my score.

---

> > > ### Author Response · Authors · 2024-08-09
> > >
> > > Thank you for your comprehensive feedback and insightful review. Your observations have been extremely beneficial in refining our manuscript. We are deeply appreciative of your recognition of our theoretical contributions and our responses to the critiques. As further detailed in the second point of our response, the methodologies we introduce are distinct from LGD-MC and DSG in their motivations and methodology, presenting complementing strategies. The experiments (Table 1 of the PDF in the global response) further indicate that the proposed methods can augment the performance of these existing works. Once again, we express our sincere thanks for your valuable comments and prompt reply.

---

### Official Review · Reviewer_wPsc · 2024-07-12

**Soundness:** 3
**Presentation:** 3
**Contribution:** 3
**Rating:** 7
**Confidence:** 3

**Summary:**

The paper studies training-free loss-based diffusion guidance. in comparison to classifier-based guidance. First, the paper examines several drawbacks of loss-based guidance, including (1) while successful minimization of the loss can be achieved, it does not guarantee successful guidance, (2) loss-based gradients can be misaligned with the desired guidance, in contrast to the time-dependent classifier-based gradients with improved smoothness properties, (3) the superior smoothness of classifier-based guidance also means that loss-based guidance takes longer to converge, incurring more function estimations.

In response to those drawbacks, the authors propose two techniques to improve loss-based guidance: (1) random augmentations to improve its smoothness, and (2) Polyak step size to further improve robustness to misalignment between initialization and the specified condition for generation. Three applications are considered for evaluation, where the proposed technique is implemented on top of FreeDoM.

**Strengths:**

- Offers theoretical results explaining the mechanisms of loss-based guidance
- Demonstrates the drawbacks of loss-based guidance, in terms of potential misalignments between its gradients and desired guidance directions, and how the smoothness properties of the employed guidance impact the rate of convergence.
- Proposes two techniques to ameliorate loss-based guidance, as demonstrated on a number of applications.

**Weaknesses:**

Nothing major, though the presentation of Section 3 leaves something to be desired. I'd need to see it after some polishing before I can be satisfied that I fully understand what's going on. See the comments below.

**Questions:**

Technical:
========
- Section 3.2 and elsewhere:
  - Recommend to use terms along the lines of "misaligned gradients" rather than "adversarial gradient."
    - I can see it helps to leverage earlier insights from adversarial robustness, e.g., regarding the role of Lipschitz constants, but I worry that there's no benefit to confusing the behavior in question with adversarial robustness considerations. If anything, those gradients were not chosen to maximize deviations w.r.t. any particular objective, but just happen to be misaligned.
    - This suggestion seems to match the narrative on L166-168.
    - Not sure if *adversarial* on L258 is related to this discussion, but just pointing it out.
  - I'm not sure what's exactly meant by "non-Lipschitz" functions
    - A function that's, say, 200-Lipschitz or only 2-Lipschitz is still not 1-Lipschitz.
    - It seems that Proposition 3.2 only needs the upper bound on $\ell$, but need not make any statements about its smoothness.
    - Similarly, L149 can simply state that adding Guassian noise improves smoothness, which is pretty intuitive.
    - It would help to double-check if Proposition 3.2 follows from known standard results, e.g., randomized smoothing which seems to be what $\hat{\ell}$ is about. See Assumption A in
      - Duchi, John C., Peter L. Bartlett, and Martin J. Wainwright. "Randomized smoothing for stochastic optimization." SIAM Journal on Optimization 22, no. 2 (2012): 674-701.
    - Same goes for Proposition 4.1.
    - is it true that Proposition 3.2 is a special case of Proposition 4.1?
- L253: Is there a way to quantify this mismatch between the condition and the input?
- I can't readily place where time travel factors into the presentation and/or the experiments. Is it only mentioned in passing? Is the related analysis in the appendix only included as a bonus?

Presentation:
==========
Section 1:
  - L43: Please cite Appendix E in [25] specifically, since the paper may otherwise seem to be concerned with a different problem altogether, i.e., offline reinforcement learning.
Section 2:
  - Suggest to flatten the section by using bold headers as used later. (Usually looks better than an absent opening paragraph.)
  - Section 2.1 could use some citations, also to clarify which formulation or model is adopted in this paper, and whether the contributions apply to other models as well.
  - L84: Please fix `fastRCNN` and include a citation.
Section 3:
  - Proposition 3.1
    - Recommend to at least provide an intuitive definition of PL. Otherwise, I'd recommend to present an *(informal)* version of the statement that's optimized for readability, deferring the formal version in its entirety to the appendix.
  - Figure 1: would help to explain that a wrong image was obtained in (b).
  - L132: Seems like a good place to start a new paragraph at "Furthermore"
  - L172: Recommend to cite a textbook.
  - L179: Promposition 3.3
    - Is it necessary to reuse the symbol $g$?
    - $g(x_t, t)$ is L-Lipschitz w.r.t. which parameter?
    - It is not clear how the discretization error is defined here. Some context would help.
    - Is it necessary to use the symbol $h_{\text{max}}$? It seems to have nothing to do with the function $h(x_t, t)$.
    - Is $O(h_{\text{max}} + Lh_{\text{max}})$ the same as $O((L+1)h_{\text{max}})$?
    - After a second reading, it's still not clear to me how the discretization error relates to the rate of convergence.
Section 4:
  - Algorithm 1 & 2: Suggest to highlight Tweedie's formula
  - L189: Please include a citation for the Polyak step size.
  - L191: Is it really the case that the analysis presented "completes the picture" for improving training-free guidance? If so, it would help to describe what it takes to complete the picture, then explain how the analysis answers to that. It would also help to highlight any remaining gaps or extensions for future work.
  - L206: Recommend to give names to those integrations defining the Lipschitz bounds.

Nitpicking:
========
- Recommend to reduce redundancy:
  - L1: Adding additional control -> Adding / Controlling / Guiding
  - L19: the generation / the synthesis, as well as the creation -> generation
  - L32: zero-shot generalization to novel conditions -> zero-shot generalization / generalization to novel conditions
  - L38: employ pre-trained networks, designed for clean images [..] checkpoints for these networks pretrained on clean images are -> employ networks pretrained on clean images [..] pretrained networks are
- Recommend to improve consistency in terminology and notation:
  - pre-trained and pretrained
  - clarifies the mystery and resolves the mystery
- Suggest to favor technical terms:
  - "mystery" of guidance weights -> role of guidance weights, principles for choosing guidance weights, intricate interplay between guidance weights and the guidance function and time.
  - L190-191: "trick" -> approach / algorithm / heuristic
- L99: Despite its intuitive -> Despite being intuitive
- L100: considers -> consider
- L100: with Gaussian dist. -> with a Gaussian dist.
- L102: for one-dimension dist. -> for a one-dimensional dist.
- L130: referred as -> referred to as
- L195: incorporating them into -> passing them to
- L212: emerges -> emerge
- L213: efficiency -> efficacy?

**Limitations:**

Although the authors seem to have aimed to complete the picture for training-free loss-based guidance, covering both its drawbacks and potential improvement strategies, this picture is still not clear to me.

For example, it's not clear what's best possible under reasonable conditions on the loss employed for guidance, e.g., how it compares to the (time-dependent) classifier models that can otherwise be employed. I suspect this has something to do with the limitations of time travel. It would be nice to attempt to show some lower-bounds on the errors incurred through time travel.

---

> ### Author Rebuttal · Authors · 2024-08-07
>
> We sincerely thank the reviewer for your insightful comments and recognition of this work, especially acknowledging our theoretical contributions.
>
> > Misaligned gradient.
>
> Thank you for suggesting a better name. We agree that the misaligned better captures the behavior and will modify it in the later versions of the manuscript.
>
> > I'm not sure ... which is pretty intuitive.
>
> We agree that the statement pointed out by the reviewer is more appropriate and will change the manuscript accordingly.
>
> > Proposition 3.2 and 4.1 and mentioned reference.
>
> We thank the reviewer for pointing out this important reference. Proposition 3.2 is a special case of Proposition 4.1 and Proposition 4.1 follows the proofs in Appendix E of the mentioned reference. We will discuss this reference in the revised manuscript.
>
> > The role of time-travel section in the appendix.
>
> Time travel is a widely adopted trick for training-free guidance, but there is no theoretical justification. In the baselines, UG, FreeDoM, and MPGD all adopted it. As a result, we include the theoretical analysis in the appendix to support when it works.
>
> > $g(x_t,t)$ is L-Lipschitz w.r.t. which parameter?
>
> We assume that $g(x_t,t)$ is L-Lipschitz w.r.t. its first parameter.
>
> > How the discretization error relates to the rate of convergence
>
> We apologize if our initial description of the discretization error was not sufficiently clear, potentially leading to confusion. To clarify, the discretization error mentioned in Proposition 3.3 refers to the discrepancy between the solution provided by DDIM and the optimal solution. In the context of diffusion, the maximum step size, denoted as $h_{\max}$, is inversely proportional to the number of steps taken. Consequently, the convergence rate can be expressed as $O((L+1)/T)$.
>
> > Is it really the case that the analysis presented "completes the picture" for improving training-free guidance?
>
> We apologize for any confusion caused by our initial explanation. In our paper, we provide theoretical justifications for the techniques used in existing training-free guidance literature, such as step size adjustments and 'time-travel.' Additionally, we have identified and analyzed key limitations. Our work aims to offer a more comprehensive understanding of the current training-free guidance approaches. We acknowledge, however, that the strategies for addressing misaligned gradients and enhancing convergence is not optimal require further development. We will ensure to clarify these points in the revised manuscript.
>
> > Presentation
>
> We sincerely thank the recommended changes. We will change the manuscript accordingly.
>
> > Nitpickings
>
> We sincerely thank the reviewer for the careful reading of the paper. We will correct these typos accordingly and have a careful check in the revised manuscript.

---

> > ### Comment · Reviewer_wPsc · 2024-08-11
> > **Follow up**
> >
> > Thanks for responding to my comments.
> >
> > I recommend to move Proposition 4.1 to the preliminaries section under a new subsection, e.g. can be titled Randomized Smoothing, with adequate citations. Proposition 3.2 should follow immediately by taking $p$ to be $\mathcal{N}(0, I)$, and can be stated as a corollary.

---

> > > ### Author Response · Authors · 2024-08-11
> > >
> > > We are immensely grateful for the time and attention you have invested in reviewing our manuscript. Your detailed comments and constructive feedback have not only highlighted areas that needed improvement but have also deepened our understanding. We want to assure you that we take your comments seriously and are currently in the process of revising our manuscript accordingly.

---

### Author Rebuttal · Authors · 2024-08-07

# Global Rebuttal

We would like to express our sincere gratitude to all the reviewers for their constructive feedback and recognition of our work. We are particularly grateful for the acknowledgment of the theoretical contributions (Reviewer wPsc, Reviewer P7k4, and Reviewer DfoY), the novelty of the identified drawbacks for existing training-free guidance (Reviewer wPsc).

First, we would like to re-emphasize the novelty and technical contributions of this work:

- Our work revisits the concept of training-free guidance from an optimization perspective, which is a novel approach that yields several insights: (1) it provides the first guarantee that generated samples from training-free guidance will have a low guidance loss (Proposition 3.1); (2) it is the first to demonstrate that training-free guidance can suffer from adversarial gradients and slow convergence (Propositions 3.2 and 3.3); (3) it offers the first theoretical support for 'time-travel', a commonly employed technique in baseline models (Proposition C.3).


- We introduce random augmentation and the Polyak step size as solutions to mitigate the issues of adversarial gradients and slow convergence, respectively. These solutions are orthogonal to previous work, allowing them to be integrated with methods such as FreeDoM, MPGD, and LGD-MC to enhance performance. Our proposed solutions have shown significant improvements over these methods and even surpass some training-based approaches.


Based on reviewers' constructive feedback, we have added more baselines and ablation studies. The main points of our rebuttal include:
- We have included additional training-free methods, DSG [A] and SAG [B], as well as the training-based method PPAP [C], in Table 1 of the PDF, addressing Weakness 1 and 2 mentioned by Reviewer DfoY and Weakness 1 by Reviewer P7k4.
- We have applied the proposed techniques to LGC-MC and MPGD-Z to demonstrate their effectiveness, as shown in Table 1 of the PDF, in response to Question 1 from Reviewer P7k4.
- We have conducted qualitative experiments to illustrate the issues of adversarial gradients and convergence in training-free guidance, which are presented in Tables 3 and 4 of the PDF, addressing Weakness 3 highlighted by Reviewer uPWj.


These changes will be integrated into the revised manuscript and we will carefully correct the typos in the revised manuscript. Please don't hesitate to let us know of any additional comments on the manuscript or the changes.

[A] Guidance with spherical gaussian constraint for conditional diffusion. arXiv:2402.03201.

[B] Towards Accurate Guided Diffusion Sampling through Symplectic Adjoint Method. arXiv:2312.12030.

[C] Towards Practical Plug-and-Play Diffusion Models, CVPR 2023.

---

### Decision · Program_Chairs · 2024-09-25

**Decision:**

Accept (poster)

**Comment:**

The reviewers and I are in agreement that the paper presents a well-isolated study and contribution on using augmentations and Polyak step sizes for improving Training-free Loss-based Diffusion Guidance.